# Beyond Blanket Masking: Examining Granularity for Privacy Protection in Images Captured by Blind and Low Vision Users

**Jeffri Murrugarra-LLerena**[1*]   **Haoran Niu**[2*]   **K. Suzanne Barber**[2]
**Hal Daumé III**[3]   **Yang Trista Cao**[2†]   **Paola Cascante-Bonilla**[1,3†]
[1] Stony Brook University   [2] University of Texas at Austin   [3] University of Maryland

## Abstract

As visual assistant systems powered by visual language models (VLMs) become more prevalent, concerns over user privacy have grown, particularly for blind and low vision users who may unknowingly capture personal private information in their images. Existing privacy protection methods rely on coarse-grained segmentation, which uniformly masks entire private objects, often at the cost of usability. In this work, we propose `FiG-Priv`, a *fine-grained privacy protection framework* that selectively masks only high-risk private information while preserving low-risk information. Our approach integrates fine-grained segmentation with a data-driven risk scoring mechanism. By leveraging a more nuanced understanding of privacy risk, our method enables more effective protection without unnecessarily restricting users' access to critical information. We evaluate our framework using the BIV-Priv-Seg dataset and show that `FiG-Priv` preserves $+26\%$ of image content, enhancing the ability of VLMs to provide useful responses by 11% and identify the image content by 45%, while ensuring privacy protection. Project Page: https://artcs1.github.io/VLMPrivacy/

## 1   Introduction

Visual assistant systems, powered by Visual Language Models (VLMs), help blind and low vision (BLV) users obtain instant answers to daily visual questions. Several such systems have been developed, including BeMyAI, FindMyThings, and SeeingAI. While the privacy policies of these applications advise users against capturing personally identifiable information (PII) in the images they upload, it is often unavoidable for blind users to unintentionally include private objects in their questions (Gurari et al., 2019). This may involve private documents, identification cards, or financial information. Additionally, some users may need to ask about personal items, further raising privacy concerns. Given these risks, it is essential to address the privacy implications of visual assistant systems and explore solutions to enhance user privacy protection.

Several datasets and methods have been proposed to identify and remove private objects in images captured by blind users (Tseng et al., 2025; Gurari et al., 2019). While these approaches have laid important groundwork, they typically rely on coarse-grained segmentation—detecting entire private objects and treating them as equally sensitive—thereby overlooking the varying degrees of privacy risk associated with different types of information. To build upon previous efforts, in this work, we introduce `FiG-Priv`, a *fine-grained privacy protection framework* that selectively masks only high-risk information within private objects while preserving low-risk information.

Not all private objects pose the same level of risk. The exposure of certain personal information, such as credit card numbers and home addresses, can lead to severe financial harm. In contrast, the disclosure of some personal items, such as pregnancy test results,

---

*Equal contribution.   †These authors jointly supervised this work.

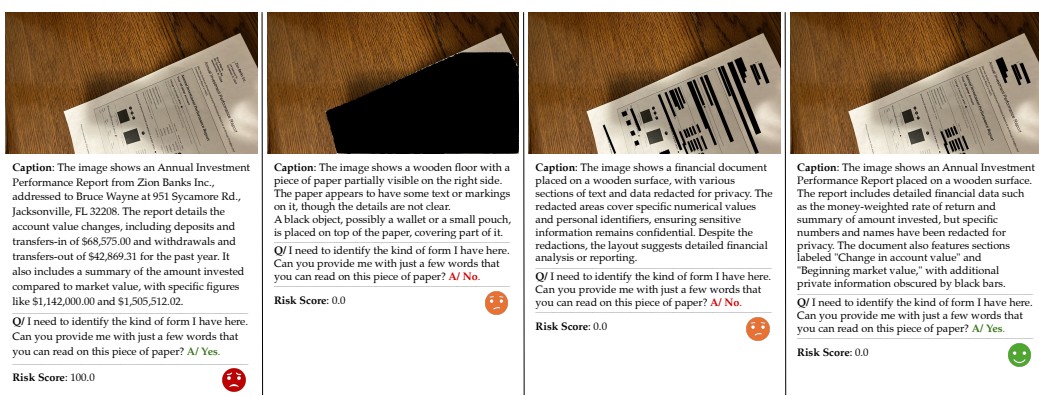

Figure 1: We show the impact of masking private objects in images taken by BLV users across levels: (a) all private information exposed, (b) the entire object masked, (c) all fine-grained detected text masked, (d) only high-risk PII masked. We show the outputs of a VLM (Caption and Question/Answer) based on the image, along with the *risk score* for each case.

may primarily cause emotional distress while posing minimal financial risk.[1] Additionally, not all information within a private object is inherently risky. Take Figure 1 as an example, blind users may ask questions about a financial statement, which is considered a private object, but their questions may pertain only to low-risk information, such as the form type or the customer service phone number. This information does not pose the same risk as other private information like the account number, account holder's name, or their personal identification number. Masking the entire statement could unnecessarily limit the model's ability to accurately answer users' questions.

To address this issue, we propose the `FiG-Priv` framework, which combines fine-grained segmentation with a data-driven risk scoring mechanism. Specifically, `FiG-Priv` incorporates a privacy risk scoring system based on The University of Texas Center for Identity's (UTCID) Identity Ecosystem graphs (Niu & Barber, 2025) and Identity Threat Assessment and Prediction (ITAP) dataset (Zaiss et al., 2019), which comprises real-world news reports of identity theft incidents (Figure 2a). The risk score reflects the relative severity and likelihood of downstream harm associated with the exposure of each type of personally identifiable information (PII). The framework also employs multiple large-scale VLMs in a multi-agent collaboration system to identify and segment high-risk PII within user-taken images (Figure 2b). Finally, we integrate the computed risk scores with the pseudo-labels generated by the multi-agent system to estimate the overall privacy risk of an image, based on the granularity of its masking, as illustrated in Figure 1.

We evaluate our framework on the BIV-Priv-Seg dataset (Tseng et al., 2025), which contains images of 16 private objects captured by BLV users. Our evaluation focuses on both the *effectiveness* and *utility* of the information masking performed by the `FiG-Priv` framework. For effectiveness, we manually annotate the masked images to assess how well the high-risk PII is concealed. For utility, we evaluate improvements in VLM performance on visual question answering tasks, enabled by preserving more low-risk visual content through our fine-grained masking approach. Our results demonstrate that `FiG-Priv` effectively protects high-risk PII, receiving an average rating of "mostly proper" by human evaluators—comparable to the rating for full-object masking. We also show that `FiG-Priv` preserves 26% more image content than full-object masking. With more visible content, a VLM can provide more useful responses with improvements of up to 11%, and better identify what elements are present in the image, achieving accuracy gains of up to 45%.

Here is a summary of our contributions: i) We propose a novel fine-grained privacy protection framework that balances privacy preservation with the model's ability to provide accurate answers for BLV users. ii) We introduce a data-driven risk scoring mechanism informed by real-world fraud and identity theft cases, enabling a more nuanced assessment

---

[1]Although this work focuses on financial risk, we acknowledge the significant impact of emotional distress on users. See the discussion section for further considerations on emotional harm.

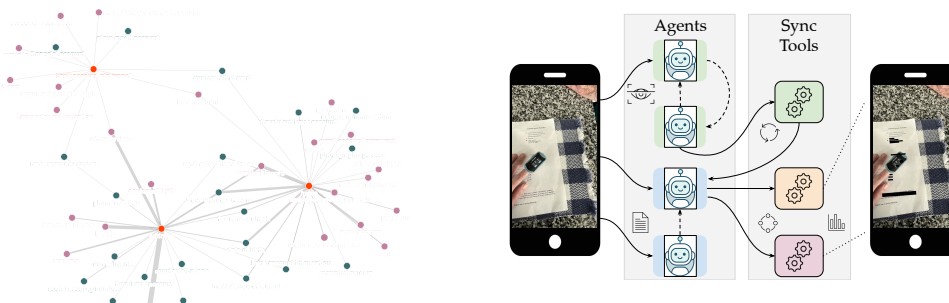

(a) UTCID Identity Ecosystem subgraph    (b) Multi-agent framework overview

Figure 2: `FiG-Priv` components. Our framework leverages multiple specialized VLM-based *agents* for tasks such as detection, segmentation, and orientation correction, while dedicated *sync*hronization *tools* coordinate their outputs for fine-grained localization.

of privacy risks in user-taken images. iii) We show that `FiG-Priv` significantly preserves the image content, which improves the ability of VLMs to provide useful responses, further enhancing their ability to identify the image content while ensuring privacy protection. A key limitation of this work is that it does not account for users' individual privacy preferences or emotional distress, relying solely on financial risk to determine risk scores. Furthermore, the framework has not yet been evaluated within real-world visual assistant applications, which we consider an important direction for future research.

## 2    Related Work

With the growing adoption of technology and AI-driven applications, privacy protection has become increasingly critical. In response to privacy challenges posed by the Internet of Things (IoT), the EU enacted the General Data Protection Regulation (GDPR), while the United States has introduced various state-level data protection laws (Bakare et al., 2024). Complementing these regulatory efforts, a range of algorithmic techniques have been proposed to support privacy preservation, including data aggregation (Lee & Chung, 2004) and user-centric privacy policy analysis frameworks (Amiri-Zarandi et al., 2020; Oltramari et al., 2018). Significant attention has also been directed toward privacy concerns in surveillance contexts (e.g., Fitwi et al., 2021; Al-Rubaie & Chang, 2019; Wang et al., 2018). Ribaric et al. (2016) present a taxonomy of privacy definitions and an overview of de-identification techniques for protecting personal information in multimedia content.

In this study, we focus on privacy preservation for images captured by BLV users within visual assistant systems. Gurari et al. (2019) first introduced and analyzed the VizWiz-Priv dataset, which contains masked images of private objects sourced from the VizWiz dataset. Sharma et al. (2023) later released the BIV-Priv dataset, which includes images of unmasked private object props taken by BLV photographers in real-world settings. To support research on identifying and removing private objects in these images, the BIV-Priv-Seg dataset was later introduced with segmentation labels for each private object (Tseng et al., 2025). Other image privacy datasets have also been proposed (e.g., Orekondy et al., 2017; Zerr et al., 2012; Spyromitros-Xioufis et al., 2016), but these typically consist of images scraped from the web and do not reflect the real-world visual content or privacy concerns relevant to visual question from BLV users. Additionally, studies have also explored BLV users' interactions with a obfuscation tool for privacy protection, highlighting users' perspectives, mental models, and preferences regarding privacy-aware interactions (Stangl et al., 2023; Zhang et al., 2024a; 2023; Alharbi et al., 2022). While prior work has provided valuable datasets and studies, technical approaches to fine-grained privacy protection for BLV users remain underexplored. To bridge this gap, we draw inspiration from recent advances in collaborative object detection and segmentation.

Recent works have explored diverse collaborative approaches for fine-grained object detection and segmentation (Hu et al., 2022; Wang et al., 2023), mostly used for autonomous

driving. Similarly, Transformer-based architectures (Xu et al., 2022a;b) leverage attention mechanisms to integrate multi-agent observations effectively. Other cooperative query-based mechanisms (Fan et al., 2024; Wang et al., 2025) have been proposed to build object-centric representations, leveraging iterative refinement through agent collaboration. In contrast, our work focuses on introducing a specialized multi-agent framework explicitly designed for privacy protection. Unlike general multi-agent architectures that only segment salient objects, `FiG-Priv` selectively identifies and masks high-risk PII in challenging images taken by BLV users, while preserving non-sensitive details.

## 3 Privacy Protection from a Granular Perspective

This section presents the core components of our proposed `FiG-Priv` framework. We begin by introducing a risk scoring mechanism based on real-world identity theft data, allowing us to quantify the risk level of various types of PII. We then introduce our approach to fine-grained private information identification, which leverages multi-agent collaboration to localize and mask only high-risk PII in user-taken images.

### 3.1 Risk Score Calculation

To calculate the risk scores associated with each type of personally identifiable information (PII), we adopt the *UTCID Identity Ecosystem graph* (Niu & Barber, 2025), which is constructed based on the *UTCID Identity Threat Assessment and Prediction (ITAP) dataset* (Zaiss et al., 2019). The dataset consists of 5,906 news stories related to identity theft, collected from a wide range of online news sources. These articles report real-world incidents involving identity theft, fraud, abuse, and exposure of PII. Each story in the dataset is annotated — if the information is present in the report — with the specific types of PII exposed (e.g., `name` and `bank account statement`), the method through which the exposure occurred (e.g., `mailbox broken`), and the resulting consequences (emotional distress or financial and property loss).

The UTCID Identity Ecosystem graph models the relationships among PII attributes based on the UTCID ITAP dataset, where each node represents a type of PII, and each directed edge indicates a causal relationship between nodes. Specifically, an edge $(u, v)$ indicates that the exposure of identity attribute $u$ may lead to the disclosure of $v$. The full Identity Ecosystem graph contains 1,718 nodes and 18,835 edges. Figure 2a illustrates a representative subgraph from the UTCID Identity Ecosystem Graph.

Based on the graph, we use the *PageRank algorithm* (Page et al., 1999) to calculate the risk scores associated with each type of PII. The algorithm was used to rank web pages in search engine results. The PageRank coefficient for each node represents the relative importance or influence of that node within a network. We use the algorithm to estimate the influence of each PII node to other nodes. A higher coefficient score indicates that the PII has greater influence, and thus poses a higher privacy risk to the user.

Specifically, we assign edge weights based on the frequency with which each causal relationship between PII types appears in the UTCID ITAP dataset; if the exposure of PII type $u$ is reported to lead to the exposure of PII type $v$ in multiple incidents, the edge $(u, v)$ receives a weight proportional to its occurrence count. We set the initial influence coefficient for each node using the total financial loss amount associated with that PII node. These initial values are then normalized to ensure that the sum of influence coefficients across all nodes equals 1. We set the convergence error tolerance of the PageRank algorithm to $1e-6$. See Appendix C and Appendix D for an analysis of the algorithms and configurations considered in our experiments.

The resulting risk scores for the PII types we focus on in this work are shown in Figure 6 [2]. The risk scores are normalized with the 90th percentile set as the maximum (100%). PII

---

[2]Risk scores are published here: `https://github.com/niu-haoran/vlm-privacy/blob/main/PII_with_PageRank_EHITS_Scores.csv`. We report risk scores only for PII types relevant to this study. The UTCID ITAP dataset is not publicly available; access was granted upon request, with permission to publish derived scores for our use case.

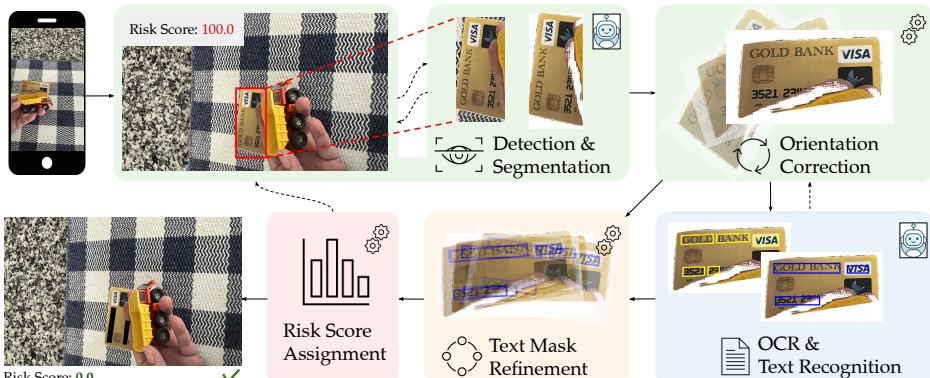

Figure 3: Multi-agent Collaboration System. Given an input image, the first agent [👁] detects, localizes, and segments the private object. Next, a second agent [↻] evaluates a set of angles to correct the object's alignment. A third set of agents processes the aligned object [▤] for OCR, text recognition, and category assignments. Then, text mask refinement [◇] is performed to map the fine-grained elements back onto the original image. Finally, the system assigns risk scores [▥] to each detected region and applies masking accordingly.

types such as `address`, `name`, and `Social Security Number` receive particularly high scores, as they are frequently associated with fraud and can lead to the exposure of additional PII, as illustrated in Figure 2a. For instance, in the news story subsection B.1, disclosing an address can result in stolen mail, which may contain other sensitive PII. See Appendix B for more news story examples. We also find that the most relevant high-risk PII primarily appears in textual form; thus, our fine-grained identification system is designed with a strong emphasis on text recognition.

## 3.2 Fine-grained Private Information Identification

Our framework aims to localize and mask only high-risk private content while preserving the remaining visual context. We implement a multi-stage pipeline composed of several agents, most of which rely on VLMs. A key challenge with photographs taken from blind and low vision users is that they are often blurry, too far away, with low light conditions, or out of focus. While prior work relies on VLMs or Optical Character Recognition (OCR) systems, these are often unable to handle these challenging cases, as shown in Figure 4 (see more comparisons in Appendix H). In contrast, our end-to-end framework leverages multiple agents that collaborate to detect and segment private objects, further processing difficult objects that contain fine-grained information, such as personal documents or credit cards. We also show a detailed example of our end-to-end framework in Figure 3.

[👁] **Detection & Segmentation** Given an input image $\mathbf{I}$, the first step in our pipeline involves identifying objects that might contain private information. We leverage a VLM-based agent to detect a candidate region within $\mathbf{I}$. This agent produces a bounding box $b = \mathcal{O}_{\text{detect}}(\mathbf{I})$. For the detected bounding box $b$, we define the corresponding cropped image $\mathbf{I}_c = \text{crop}(\mathbf{I}, b_i)$. This cropped image is then processed by a segmentation module based on a VLM, which refines the object's boundaries by producing an object-level mask $m = \mathcal{O}_{\text{segment}}(\mathbf{I}_c)$. Finally, the masked version of the cropped image is obtained by $\hat{\mathbf{I}}_c = \mathbf{I}_c \odot m$, where $\odot$ denotes element-wise multiplication, setting pixels outside the mask to 1 (i.e., white). These refined outputs serve as the basis for the next stages in our pipeline. Additional details in subsection E.1.

[↻] **Orientation Correction** Since the segmented object may be in difficult positions and orientations for text recognition, we evaluate a set of candidate rotation angles $\Theta = \{\theta_1, \theta_2, \dots\}$ to correct $\hat{\mathbf{I}}_c$. For each angle $\theta$, we rotate the object and query a VLM-based agent to assess whether the image is correctly aligned. We then select $\theta^*$ that yields the highest alignment probability and apply it to obtain the best approximately aligned image for the next step. Additional details in subsection E.2.

📄 **Text Recognition** Following orientation correction, the rotated object $R_{\theta*}$ is processed by a text-localization module that leverages two specialized agents for text extraction. Te first one, denoted as $\text{OCR}_{\text{agent}}$, outputs a set of text segments along with detailed detection results in the form of polygons. We use these polygons to further refine the orientation of the document, by computing the orientation angle of each polygon with respect to the horizontal axis. The second one, denoted as $\text{VLM}_{\text{agent}}$, is able to recognize text and outputs its detections as bounding boxes. We combine the output of both agents for the subsequent text mask refinement. Additional details in subsection E.3.

◇ **Text Mask Refinement** Here, we convert the OCR-detected polygons (*polyOCR*) and the VLM-detected bounding boxes (*boxVLM*) of $R_{\theta*}$ to polygons in the coordinate space of the original image **I**, by mapping each *boxVLM* to a four-point polygon. Next, for each point $p$ in *polyOCR* or *polyVLM*, we apply an inverse rotation to map it back to the coordinate space of the original cropped image. We then re-align these refined polygons within the full original image by adding the top-left coordinates of the detected object's bounding box. If no such reference exists, a default origin of $[0, 0]$ is assumed. Details in subsection E.4.

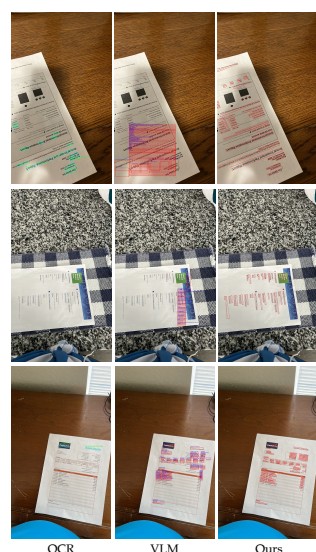

Figure 4: Comparison of fine-grained outputs. Fig-Priv accurately detects all content, while specialized OCR and VLMs fail.

📊 **Risk Score Assignment** Finally, for each detected region or text segment $t$ obtained from the combined outputs of the Text Recognition and Text Mask Refinement stages, we run a VLM-based categorization agent to predict a sub-category $s$ for the text segment $t$ based on the categories of our Risk Score Calculation (introduced in Section 3.1): $s = \mathcal{O}_{\text{VLM}}(t, c_t)$. If the predicted sub-category $s$ belongs to the high-risk set $H_r(c_t)$, the text segment is marked as high risk. Consequently, the corresponding region in the original image **I** is masked. Formally, let $\mathcal{H}_{\mathcal{R}}$ be the set of all high-risk text segments: $\mathcal{H}_{\mathcal{R}} = \{t \mid s \in H_r(c_t)\}$, and define the final masked image as $\hat{\mathbf{I}} = mask(\mathbf{I}, \mathcal{H}_{\mathcal{R}})$, where *mask* is an operation that takes the coordinates associated with each text segment in $\mathcal{H}_{\mathcal{R}}$ and obscures those regions by overlaying a solid black color. This final output ensures that only high-risk content is removed, preserving the utility and context of the remaining data.

## 4 Experiment Setup

In our evaluation, we assess the effectiveness and utility of the `FiG-Priv` framework with two research questions: 1) **RQ-PrivProt**: whether the framework effectively protects private information without full-object masking and 2) **RQ-Perf**: how well VLMs perform on VQA tasks using images processed by the framework. We describe our experimental setup in the following section. By preserving more low-risk visual content, our fine-grained masking strategy has the potential to improve usability and maintain model performance.

**Dataset:** To evaluate the `FIG-Priv` framework on realistic images captured by blind and low-vision users, we use the BIV-Priv-Seg dataset (Tseng et al., 2025). This dataset contains 1,028 images taken by blind and low-vision individuals on private objects. To ensure the images closely resemble real-world use cases while protecting photographers' privacy, all private objects in the dataset are "props" (e.g., a fake credit card instead of a real one). To our knowledge, BIV-Priv-Seg is the only public dataset that includes unmasked private objects in images taken by blind and low-vision users.

The BIV-Priv-Seg dataset includes 16 private object categories, listed in Table 1. For each category, we construct a list of fine-grained PII types and their associated risk scores, also shown in Table 1 (full list in the appendix Table 6). To build this list, we first identify

| Private Object Category | PII |
|---|---|
| bank statement | address, bank account balance, bank account number, bank account transfer amount, date of birth, ... |
| letter with address | address, name, phone number, signature |
| credit or debit card | bank card expiration date, credit card number, cvv code, debit card number, name, signature |
| bills or receipt | address, bill paid, credit card number, customer account, customer account information, date of birth, ... |
| preganancy test | - |
| pregnancy test box | - |
| mortage or investment report | account number, address, bank account number, consumer good and/or service, date of birth, ... |
| doctor prescription | address, date of birth, diagnosis, email, medication prescribed, name, personal identification number |
| empty pill bottle | address, date of birth, diagnosis, email address, name, personal identification number |
| condom with plastic bag | - |
| tattoo sleeve | biometric data |
| transcript | address, date of birth, education level, email address, name, school attended |
| business card | address, email address, employment history, employer name, name, phone number, position |
| condom box | - |
| local newspaper | - |
| medical record document | address, date of birth, diagnosis, email address, name, personal identification number |

Table 1: Private Object Categories with Associated PII Types.

synonym node terms in the Ecosystem graph for each private object (e.g., `letter with address` → `mail`). Using these terms, we extract all nodes directly connected to the private object, as they represent information that may be disclosed when the object is exposed. From these, we select nodes that represent information typically contained within the object (e.g., `mail` → `address`) to form our fine-grained PII list.

**Evaluation Setup:** To answer **RQ-PrivProt**, we randomly sample 168 images from the BIV-Priv-Seg dataset and manually annotate them to evaluate the masking quality of our `FiG-Priv` framework. For each image, we generate three masked versions representing different strategies: (1) full-object masking, (2) fine-grained masking of all information related to the private object, and (3) fine-grained masking of only high-risk PII (our framework). Each image is rated on a 5-point Likert scale, where 1 indicates completely ineffective masking (PII fully exposed) and 5 indicates perfect masking (PII fully obscured with clear, consistent masking). See Appendix J for question and scale details. Each masked image is independently annotated by two annotators. Based on these ratings, we assess how effective our framework is at protecting PII compared to the other strategies.

Additionally, to answer **RQ-Perf**, we evaluate VLM performance on the VQA task using images processed by the three masking strategies. However, since the BIV-Priv-Seg dataset does not include visual questions, we synthesize a set of questions for each image. Object recognition is the most common type of query from BLV users, so our first experiment tests the VLMs' ability to recognize the private object under different masking conditions. We prompt the model with: "Is there a [private object] in the image? Answer yes or no", where [private object] is the meta-category assigned by the Multi-agent Collaboration System. To evaluate model performance when objects appear in both the foreground and background, we also construct a secondary VQA dataset by inpainting synthetic *control objects* with known attributes and locations. These objects are sampled from the BOP-HOT3D dataset (Hodan et al., 2018; Banerjee et al., 2024), which contains 33 texture-mapped 3D objects commonly found in indoor environments. We select 8 objects for which the models show the highest confidence and randomly insert them into the images without occluding the original private object. We compute the model's likelihood of answering "yes" and report the model's accuracy under different masking strategies.

Furthermore, to evaluate VLM performance on realistic questions from BLV users, we construct a set of questions for each private object. Specifically, we manually select questions from the VizWiz dataset (Bigham et al., 2010) that can be answered without high-risk PII. For each private object, we select 2–5 relevant questions. Some objects have fewer questions due to the limited availability of queries related to low-risk information. See Appendix Table 8-18 for the full list of questions. Since we do not have ground-truth answers for these questions, we only prompt the model to indicate whether the question is answerable. We conduct our experiments on LLaVA-1.6 (Liu et al., 2024), Qwen2.5-VL (7B) (Bai et al., 2025), and Gemma-3 (4B) (Google, 2025). These models are more likely to be used in consumer-level applications, given their parameter size and inference time. Additional technical details for the multi-agent framework and components are available in Appendix E.

| Masking Type | Average Rating | Annotator Agreement |
|---|---|---|
| Object Mask | 4.2 | 91% |
| Fine-grained Mask | 4.0 | 91% |
| High-Risk Mask | 3.9 | 93% |

Table 2: Ratings and inter-annotator agreement on effectiveness of each masking strategy.

| Model | Image Content | Private Object | Control Object | $PO_{dif} \downarrow$ | $CO_{dif} \uparrow$ |
|---|---|---|---|---|---|
| LLaVA-1.6 (Liu et al., 2024) | Full Image | $0.572 \pm 0.02$ | $0.755 \pm 0.20$ | – | – |
| | Object Mask | $0.035 \pm 0.01$ | $0.816 \pm 0.20$ | -0.5357 | +0.0608 |
| | Fine-grained Mask | $0.286 \pm 0.02$ | $0.810 \pm 0.20$ | -0.2861 | +0.0548 |
| | High-Risk Mask | $0.437 \pm 0.02$ | $0.799 \pm 0.20$ | -0.1351 | +0.0438 |
| Qwen2.5-VL (Bai et al., 2025) | Full Image | $0.806 \pm 0.01$ | $0.975 \pm 0.03$ | – | – |
| | Object Mask | $0.145 \pm 0.01$ | $0.983 \pm 0.03$ | -0.6612 | +0.0082 |
| | Fine-grained Mask | $0.530 \pm 0.02$ | $0.979 \pm 0.03$ | -0.2756 | +0.0036 |
| | High-Risk Mask | $0.647 \pm 0.01$ | $0.976 \pm 0.03$ | -0.1593 | +0.0005 |
| Gemma-3 (Google, 2025) | Full Image | $0.915 \pm 0.01$ | $0.845 \pm 0.05$ | – | – |
| | Object Mask | $0.435 \pm 0.02$ | $0.967 \pm 0.02$ | -0.4804 | +0.1219 |
| | Fine-grained Mask | $0.823 \pm 0.01$ | $0.963 \pm 0.03$ | -0.0922 | +0.1179 |
| | High-Risk Mask | $0.871 \pm 0.01$ | $0.882 \pm 0.05$ | -0.0442 | +0.0369 |

Table 3: VLM Performance. Image Content indicates the masking condition applied to the image, ranging from no masking (i.e., the full image) to masking only high-risk content. Private Object refers to questions focused on the private object, while Control Object refers to questions focused on a control object placed in the image, with private objects appearing in the background. $PO_{dif}$ indicates the degradation of the Private Object after masking, and $CO_{dif}$ indicates the accuracy difference of the Control Object.

## 5 Results

**RQ-PrivProt:** To evaluate the effectiveness of high-risk masking with the FiG-Priv framework, we conduct a human annotation study to rate the quality of different masking strategies. The results are presented in Table 2. We observe that, for all the questions, annotator agreement exceeds 90%, where agreement is defined as a score difference of no more than one point on the Likert scale. In general, all masking strategies achieve "mostly proper" masking, effectively covering most private information (Likert scale = 4). Compared to Object Mask and Fine-grained Mask, which have average ratings of 4.2 and 4.0 respectively, the High-risk Mask receives a slightly lower average rating of 3.9.

From the annotation results, we identified several common issues that contribute to lower masking quality in the high-risk masking strategy. First, image quality has a significant impact. Some images are heavily blurred, making it difficult for the model to understand the context (e.g., Figure 8d). However, they are not so blurred that humans, or image restoration techniques, cannot recover the high-risk PII. Moreover, some images only partially capture the private object, which makes it difficult for the model to accurately predict its subcategory pseudo-labels. For example, in Figure 8h, a credit card is only partially visible, and the name on the card was not correctly labeled, resulting in a failure to mask it. Second, technical limitations of the framework also contribute to masking errors. Several low-quality maskings result from mask misalignment (e.g., Figure 8k). Since private objects often appear in various orientations. Though we employ an orientation agent to account for this, some cases still fall outside its handling capabilities. Finally, some high-risk PII appears within some text, and the labeling agent may fail to detect and mask this information. For instance, in Figure 8p, a name written in a letter was not masked by the framework.

**RQ-Perf:** To evaluate VLM performance across the three masking strategies, we first test object recognition in two settings: when the private object is the focus of the question, and when a control object is the focus with the private object in the background. Results are shown in Table 3. For questions targeting recognition of the private object, high-risk masking achieves accuracy scores closest to those of the unmasked (full image) condition. Compared to full-object masking, it yields an average accuracy improvement of 45% across the three VLMs. For questions where the private object appears in the background, there is no significant accuracy difference across the three masking strategies.

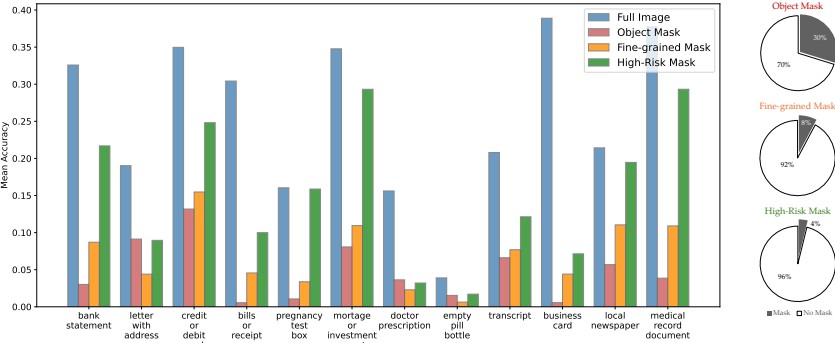

Figure 5: Left: Answerability of a model given a set of common questions on images under different masking conditions. We include all questions in the appendix Table 8-18. Right: Average masking of image content based on different conditions.

We also evaluate VLM performance on the VQA task using realistic, human-asked questions. As shown in Figure 5-Left, high-risk masking achieves an answerability rate closest to that of the full image. On average, the answerability rate increases by 11% when comparing high-risk masking to full-object masking. See Appendix Table 8–18 for a detailed breakdown of scores by object and question. We also compute the percentage of the image masked under different masking strategies. As shown in Figure 5-Right, high-risk masking preserves 26% more image content compared to full-object masking.[3] This greater preservation of visual content may explain the higher answerability rate observed with high-risk masking.

In general, we observe low answerability rates across all objects ($< 40\%$), with empty pill bottles showing the lowest rate ($< 5\%$). This may be due to poor image quality—many images are too blurry for the model to reliably detect text. In such cases, full-object masking may be preferable, as the question is unlikely to be answerable while the risk of information exposure still remains.

Among the objects, bank cards and documents (e.g., mortgage or investment reports) show relatively higher answerability rates under high-risk masking. In contrast, doctor prescriptions and business cards have lower answerability rates. For these objects, many questions relate to information that falls near the boundary of being high-risk, such as medical diagnoses or phone numbers on business cards. In our current setup, we take a conservative approach by masking all such content. However, in real-world applications, users could be given the ability to adjust masking thresholds based on their own privacy preferences and needs. Overall, our results show that VLMs are able to answer more user questions under high-risk masking, while still preserving appropriate privacy protection.

**Is an agentic approach necessary in this context?** We identify significant limitations of current state-of-the-art models in accurately recognizing and localizing information present in challenging images. We compare our method with Gemini2.5 (Comănici & et al., 2025), GPT4o (OpenAI, 2024), MistralOCR (Mistral AI, 2025), Qwen2.5-VL (Bai et al., 2025) and PaddleOCR (Yanjun et al., 2019; Cui et al., 2025). We show that our multi-agent system achieves precise content localization, enabling the context-aware masking required for effective privacy preservation while maximizing content utility, critical for BLV users, as shown in Figure 4 and Appendix H. Finally, we further discuss the computational overhead and real-time feasibility of our method in Appendix I.

## 6   Conclusion and Discussion

FiG-Priv combines a data-driven risk scoring mechanism with a multi-agent vision-language model (VLM) system to selectively mask only high-risk personally identifiable information (PII). Through our evaluation on the BIV-Priv-Seg dataset, we demonstrated

---

[3]Some objects, such as local newspapers and pregnancy test boxes, are not classified as high-risk; therefore, high-risk masking produces the same score as the full image for those cases.

that a granular, risk-aware approach to privacy protection achieves a more effective balance between safeguarding user privacy and maintaining the usability of visual assistant systems.

The problem of privacy protection is complex – we must strike a careful balance between protecting personal information and maintaining the usability of AI applications. This trade-off is especially important for BLV users, who increasingly use visual assistant systems to access visual information in their daily lives. Studies have shown that BLV users are enthusiastic about AI technologies (e.g., Khan et al., 2020). Thus requiring users to compromise on either privacy or usability creates an unfair burden and increases the risk of their PII exposure. Therefore, we hope our study contributes toward building AI systems that support both privacy and accessibility.

This study also raises several important considerations for future research. As we continue to explore privacy-preserving visual assistance, it is increasingly important to incorporate BLV users directly into the design and evaluation of such systems. The current implementation uses a basic redaction method –black square masking– which, while operationally simple, may not adequately account for user experience or support optimal system functionality. Prior research Zhang et al. (2024a) has demonstrated that the choice of redaction technique, such as blurring or replacement, can influence both user perception and the performance of visual-language models. Furthermore, the present system applies uniform masking to all instances of PII with non-zero risk scores, without accounting for individual variability in privacy preferences. Some users may accept greater information exposure in exchange for improved task performance, while others may favor more information protection. Enabling adjustable privacy configurations and assessing their impact in real-world assistive contexts may offer a more user-aligned and practical approach to system deployment.

Finally, the risk scoring framework in this study relies solely on estimated financial losses from identity theft, offering a limited perspective on privacy risk. Our risk scoring mechanism is currently based solely on estimated financial losses from identity theft, which limits its scope. Emotional and psychological impacts, such as embarrassment or anxiety, are not captured. Future work could explore ways to incorporate these emotional dimensions into risk assessments, offering a more holistic approach to privacy protection. It is also important to recognize potential biases in the current scoring model. Temporal bias may arise as threat patterns and identity theft tactics evolve, making historical data less representative of present risks. Reporting bias can further affect the data, since incidents of identity theft are not uniformly reported or documented. Additionally, demographic and cultural factors shape both the occurrence and perception of privacy violations, leading to disparities in risk assessment across different populations and contexts. Addressing these biases is essential for creating more equitable and contextually sensitive risk assessment mechanisms.

## 7   Ethical Considerations and Limitations

This study is intended to support privacy protection, not to facilitate the extraction of PII from images. In particular, we caution against the misuse of our methods or datasets for identifying PII in public images. This is especially important in the context of visual assistant applications, which often handle sensitive visual data from BLV users.

Despite its contributions, this study has certain limitations that should be acknowledged and addressed in future research. One major limitation is that our framework cannot appropriately handle all biometric data, such as tattoos. The importance of biometric data is increasing, as it can uniquely identify individuals and may carry sensitive personal, cultural, or medical significance. Orekondy et al. (2017) proposed algorithms for detecting certain personal attributes in images, but future work should advance biometric data protection, particularly in the context of realistic images captured within visual assistant systems.

Additionally, our multi-agent collaboration system has technical limitations. It struggles to reliably identify certain types of textual information, such as signatures, QR codes, and barcodes. Since such information can carry sensitive personal or financial data, incorporating specialized detectors or training on targeted datasets to improve the system's ability to recognize and mask these items is an important direction for future work.

## Acknowledgments

This material is based upon work supported by the NSF under Grant No. 2229885 (NSF Institute for Trustworthy AI in Law and Society, TRAILS). We thank the anonymous reviewers for their thoughtful and thorough feedback. We used GPT for writing support and Ai2 PaperFinder for assistance with the literature review.

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

## A  Risk Score Details

Figure 6 shows the risk scores we assign to the listed PII attributes. We use the PageRank algorithm to calculate the scores. Each PII attribute has two different risk scores. For the two types of risk scores, one is generated from the PageRank algorithm with edge weight being the occurrence frequency of the edge. The other risk score is calculated based on the PageRank algorithm with edge weight being the total loss amount associated with the edge.

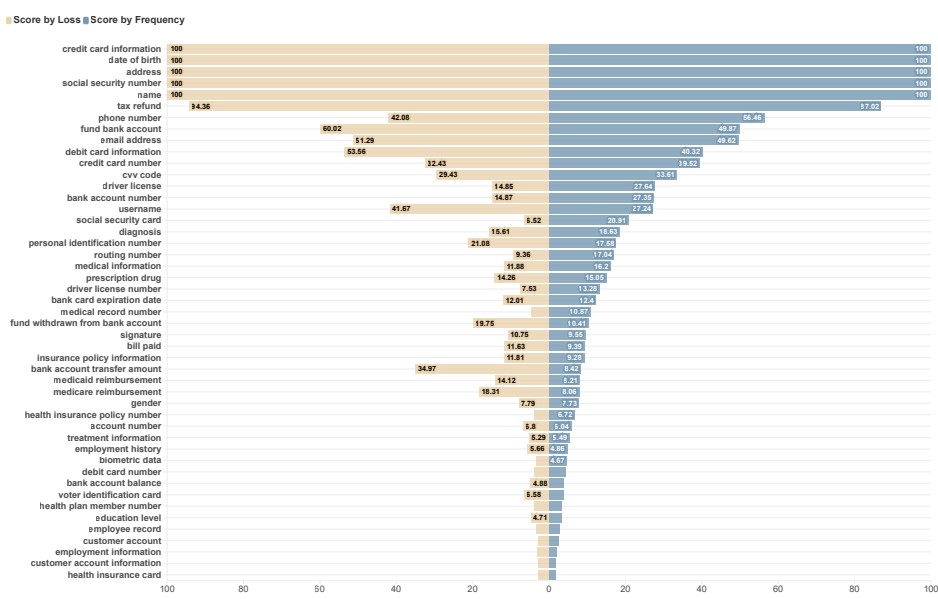

Figure 6: Risk score associated with each PII type

## B  News Story Examples from the ITAP Dataset

### B.1  News Story 1

**Source**: Mail with PII Stolen Mailbox(es) Broken Into
**Target**: Check Information Altered, Check(s) Deposited, Monetary Amount, Stolen Check
**News summary**: An identity thief broke a victim's mailbox and get the victim's mail. The mail contains the victim's check information and other related information to get the cash out. The mail causes the disclosure of check information and other monetary transaction related personal information.

### B.2  News Story 2

**Source**: Credit Card Application(s) Submitted, Date of Birth, Name, PII Distributed via Email, PII Stolen, Server(s) Accessed without Authorization, Social Security Number
**Target**: Consumer Goods and/or Services Purchased, Counterfeit Credit Card(s) Created
**News summary**: A company's server is accessed by an identity thief. The thief got customers' email addresses and sent phishing emails. The customer replied to the email with important PII including date of birth and name. The thief submitted credit card applications using victims' PII. The credit card application was approved. The thief uses the approved cards to make transactions.

### B.3  News Story 3

**Source**: Email Address, Name, Phishing Email Sent
**Target**: Name, Social Security Number, W-2 Form Information

**News summary**: An identity thief got a victim's email address. The theft sent phishing emails to make the victim send PII including name, Social Security Number and W-2 form information.

### B.4 News Story 4

**Source**: Name, Social Security Number, Social Security Number(s) Stolen
**Target**: Fraudulent Bank Account(s) Opened, Fraudulent Loan Taken Out
**News summary**: A victim's name and Social Security Number are stolen by an identity theif. The thief uses the PII information to open a fraudulent bank account under the name of the victim and take the loan out from the fraudulent bank account.

### B.5 News Story 5

**Source**: Employee Credentials Stolen, Malware Injected, Password, Username
**Target**: Bank Account Information, Bank Account(s) Compromised, Name, Payroll System Breached
**News summary**: An identity thief used malware to get a company employee's credentials. The thief got customers' password and username information. Using the password and username, the thief eventually got customers' bank account information, name and payroll information.

## C PageRank Algorithm

The PageRank algorithm is shown as Algorithm 1.

---
**Algorithm 1** PageRank Algorithm

---
**Input:** $G : \langle V, E \rangle$
**Parameters:** damping factor: $d \in (0, 1)$; convergence threshold: $\epsilon$
$V \leftarrow G.nodes$
$E \leftarrow G.edges$
for $v$ in $V$:
    $PageRankCoefficient[v] = \frac{1}{len(G.nodes)}$
while the maximum change in *PageRankCoefficient* of across all nodes is greater than $\epsilon$:
    for $u$ in $V$:
        $PageRankCoefficient[u] = \sum_{(v,u) \in E} d * PageRankCoefficient[v] * \frac{1}{degree(v)}$

---

| E-HITS 
 PageRank | hub - freq | authority - freq | hub+authority - freq |
|---|---|---|---|
| PageRank - freq | 0.65 | 0.79 | 0.79 |

Table 4: Spearman Correlations Between Coefficients Generated by the Two Algorithms (Using Occurrence Frequency as the UTCID Identity Ecosystem Graph Weights).

| E-HITS 
 PageRank | hub - loss | authority - loss | hub+authority - loss |
|---|---|---|---|
| PageRank - loss | 0.66 | 0.81 | 0.81 |

Table 5: Spearman Correlations Between Coefficients Generated by the Two Algorithms (Using Financial Loss Amounts as the UTCID Identity Ecosystem Graph Weights).

---

**Algorithm 2** E-HITS Algorithm

---

**Start with the graph:**$G:\langle V, E \rangle$
$V \leftarrow G.nodes$
$E \leftarrow G.edges$
$maxInDegree \leftarrow max(G.inDegree)$ # Get the maximum in degree of graph $G$.
$maxOutDegree \leftarrow max(G.outDegree)$ # Get the maximum out degree of graph $G$.
for $v$ in $V$: # Initialization.
    $hub[v] = 1$
    $authority[v] = 1$
    $risk[v] = 1$
while $hub$ and $authority$ not convergent:
    for $v$ in $V$:
        $hub[v] = \sum_{(v,u) \in E} authority[u] * \frac{weight(v,u)}{maxOutDegree}$
        $authority[v] = \sum_{(u,v) \in E} hub[u] * \frac{weight(u,v)}{maxInDegree}$
        $Normalize(hub)$
        $Normalize(authority)$
        $risk[v] = hub[v] + authority[v]$
return $hub, authority, risk$

---

## D  E-HITS Algorithm

Instead of the PageRank algorithm, we can also use the Edge-weighting Hyperlink-introduced Topic Search Algorithm (*E-HITS algorithm*) Hoa & Ha (2017) as shown in Algorithm 2. Hyperlink-introduced Topic Search (HITS) algorithm is commonly used to identify important nodes in directed graphs, such as web page hyperlink connection graphs and social networks. Building on the original HITS algorithm, the E-HITS algorithm incorporates the actual edge weights while still generates hub and authority scores for each node. We use the E-HITS algorithm to estimate the influence of each PII node within the Identity Ecosystem Graph. Specifically, for each node, we calculate the sum of its hub and authority scores. A higher combined score indicates that the PII has greater influence and, therefore, poses a higher privacy risk to user.

The risk scores generated by the E-HITS algorithm are highly correlated with those generated by the PageRank algorithm. Table 4 and 5 present the Spearman correlations (Spearman, 1961) between the coefficients (i.e., risk scores) generated by the PageRank and E-HITS algorithms, respectively. Thus, in this work, we picked to use the PageRank algorithm score as our risk scores.

## E  Multi-agent Framework Components

We provide a detailed description of each agent below.

### E.1  Localization Agents ($\mathcal{O}_{\textbf{detect}}$ & $\mathcal{O}_{\textbf{segment}}$)

These agents are responsible for providing a region proposal and detailed mask of potential private objects respectively. We define each model below.

- $\mathcal{O}_{detect}$, is the starting point for our pipeline. Large-scale VLMs with good grounding capability can be a good choice for this agent, as they provide good coarse detections for potential private objects. We use Qwen2.5-VL (Bai et al., 2025) 72B, a large-scale VLM. We use the prompt: `"Locate [private object] in the image and output in JSON format."` along with the image.

- $\mathcal{O}_{segment}$, refines the coarse detection. By segmenting the object of interest, we aim to remove background distractors. We use EVF-SAM (Zhang et al., 2024b), a lightweight version of SAM (Kirillov et al., 2023) with text-prompt capabilities.

Along with our coarser detection, we query the model with a prompt related to the *private object* (e.g., document for paper document). When coarser detection is not found, we use the whole image.

## E.2 Orientation Agent ($\mathcal{O}_{\text{orientation}}$)

This agent determines the best orientation of a *private object* that facilitates text recognition. For each rotated images, we prompt Qwen2.5-VL (Bai et al., 2025) 7B with "Is the text in this document readable (top down, left to right)? Answer yes or no" and select the one that maximizes the yes option. Given that the segmented objects may be in difficult positions and orientations for each segmented object $\hat{\mathbf{I}}_c$, we address potential skew or misalignment by evaluating a set of candidate rotation angles $\Theta = \{\theta_1, \theta_2, \ldots, \theta_k\}$ to correct $\hat{\mathbf{I}}_c$. For each candidate angle $\theta \in \Theta$, we rotate the image

$$R_\theta = \text{rotate}(\hat{\mathbf{I}}_c, \theta)$$

and query a VLM-based agent to determine the likelihood of the image being correctly aligned. Finally, we select the angle $\theta^*$ that maximizes this probability, and the segmented object $\hat{\mathbf{I}}_c$ is rotated by $\theta^*$ to obtain the best approximately aligned image for the next step.

## E.3 Text Recognition Agents ($\mathcal{O}_{\text{OCR}}$ & $\mathcal{O}_{\text{VLM}}$)

These agents produce independent outputs that contain fine-grained locations and corresponding text.

- $\mathcal{O}_{\text{OCR}}$, we employ PaddleOCR [4] using their implementation of the PGNet (Wang et al., 2021) method that was trained in the TotalText (Ch'ng et al., 2020) dataset.
- $\mathcal{O}_{\text{VLM}}$, we use Qwen2.5-VL (Bai et al., 2025) 72B and prompt the model with "Locate all text (bbox coordinates). Include all readable and blury text".

The first agent, denoted as $\text{OCR}_{\text{agent}}$, is a large-scale model specialized for OCR. It outputs a set of text segments along with detailed detection results in the form of polygons:

$$\mathcal{O}_{\text{OCR}}(R_{\theta^*}) = \{(t_j^{(1)}, p_j^{(1)}) \mid j = 1, \ldots, N_1\},$$

where $t_j^{(1)}$ is a recognized text segment and $p_j^{(1)}$ is the corresponding polygon (*polyOCR*). Then, we estimate the degree to which each detected polygon is horizontally aligned. For each polygon $p \in \{p_j^{(1)}\}$, we compute its orientation angle $\phi_p$ as the angle between its primary axis and the horizontal axis. Based on $\phi_p$, we apply an additional correction by rotating the corresponding cropped image by $-\phi_p$ before further processing. This additional step ensures that the image is optimally aligned for text detection by the second agent.

Denoted as $\text{VLM}_{\text{agent}}$, the second agent is a large-scale visual language model able to recognize text and outputs its detections as bounding boxes:

$$\mathcal{O}_{\text{VLM}}(R_{\theta^*}) = \{(t_k^{(2)}, p_k^{(2)}) \mid k = 1, \ldots, N_2\},$$

where $t_k^{(2)}$ is a text segment and $p_k^{(2)}$ is the corresponding bounding box (*boxVLM*). These two approaches are complementary: the OCR-specialized agent yields precise polygonal detections, while the VLM-based agent contributes robust text localization even under challenging image conditions. The combined output of the text recognition stage is given by:

$$\mathcal{O}(R_{\theta^*}) = \mathcal{O}_{\text{OCR}}(R_{\theta^*}) \cup \mathcal{O}_{\text{VLM}}(R_{\theta^*}),$$

providing a comprehensive set of recognized text segments and their associated spatial representations. This output forms the basis for subsequent text mask refinement.

---

[4]https://github.com/PaddlePaddle/PaddleOCR/blob/main/README_en.md

### E.4 Text Mask Refinement

In this stage, we convert the OCR-detected polygons (*polyOCR*) and the VLM-detected bounding boxes (*boxVLM*) of $R_{\theta*}$ to polygons in the coordinate space of the original image **I** as follows: first, we map each *boxVLM*, which is provided in the format $[x_{\text{top}}, y_{\text{top}}, x_{\text{bottom}}, y_{\text{bottom}}]$ (i.e., top-left and bottom-right coordinates), to a four-point polygon, denoted as *polyVLM*:

$$polyVLM = \big[(x_{\text{top}}, y_{\text{top}}), (x_{\text{top}}, y_{\text{bottom}}), (x_{\text{bottom}}, y_{\text{top}}), (x_{\text{bottom}}, y_{\text{bottom}})\big].$$

Next, for each point $p$ in *polyOCR* or *polyVLM*, we apply an inverse rotation to map the point back to the coordinate space of the original cropped image. Let $c$ denote the center of the rotated image, and let $R_{-\theta}$ be the rotation matrix corresponding to an angle $-\theta$ (with $\theta$ being the angle used in the orientation correction). For each point $p = (x, y)$, the transformed point $p'$ is computed as:

$$p' = R_{-\theta}(p - c) + c$$

so that all resulting polygons are aligned with their corresponding cropped objects.

Finally, we re-align these refined polygons within the full original image by adding the top-left coordinates of the detected object's bounding box. If no such reference exists, a default origin of $[0, 0]$ is assumed. Thus, for each refined point $p'$, its final coordinate in the original image is given by:

$$p'' = p' + (x_{\text{tl}}, y_{\text{tl}}).$$

This process ensures that the detected masks align with the positions of the localized text in the original image.

**Labeling Agent ($\mathcal{O}_{\textbf{labeling}}$):** This agent acts as a Named Entity Recognition. Given an image, for each detected text we query the Qwen2.5-VL (Bai et al., 2025) 72B with "Based on the image, classify this text: [strs] using these categories: [private_object_categories]. Output only one category.".

## F  Private Object Categories

For each item in the private object category, we find its corresponding synonym PII nodes from the identity Ecosystem graph. Some PII synonym nodes contain other PII attributes that also appear in the identity Ecosystem graph as nodes. For instance, the PII node "mail" has "address" and "name" associated with it. If we lose a mail, it usually means the address and the sender and recipient's names may be disclosed. Sometimes phone numbers and signatures are also included in a mail. Therefore, for the private object "letter with address" in the first column, we have "mail" as synonym PII nodes and the PII information contained in "mail" are: address, name, phone number and signature.

## G  Visual Results

In this section, we show visual results from FiG-Priv framework. Figure 7 and Figure 8 present for favorable and non-favorable scenarios respectively. In particular, we display from left to right: the Full image, the Object Mask within the image, the Fine-grained mask within the image, and the High-Risk mask within the image.

Figure 7 show several examples where our framework can detect high-risk objects even under challenging scenarios. Figure 7a-7d show an image with a high amount of fine-grained information which the framework successfully capture and produce also a High-Risk mask. Figure 7e-7h show a clear example where the risk masks were selected from all the fine-grained masks. Figure 7i-7p present document with some degree of rotation where FiG-Priv correctly aligned the fine-grained detection and use them to label and output High-Risk masks.

| Private Object Category | Synonym Nodes | PII |
|---|---|---|
| bank statement | bank account statement, bank account information | address, bank account balance, bank account number, bank account transfer amount, date of birth, email address, employment history, medicaid reimbursement, medical information, medicare reimbursement, name, personal identification number, routing number, social security number, tax refund, username |
| letter with address | mail | address, name, phone number, signature |
| credit or debit card | bank card, credit card, credit card information, debit card, debit card information | bank card expiration date, credit card number, cvv code, debit card number, name, signature |
| bills or receipt | billing information, billing statement, finance invoice and receipt, invoice and/or receipt | address, bill paid, credit card number, customer account, customer account information, date of birth, email address, medical information, name, personal identification number, signature |
| preganancy test | diagnosis, medical record | - |
| pregnancy test box | - | - |
| mortage or investment report | financial information, financial statement, loan application, loan information, loan statement, mortgage document | account number, address, bank account number, date of birth, email address, employee record, fund bank account, fund withdrawn from bank account, name, social security number |
| doctor prescription | medical information, medical prescription, medication prescription | address, date of birth, diagnosis, email address, medication prescribed, name, personal identification number |
| empty pill bottle | medical information, medical record | address, date of birth, diagnosis, email address, name, personal identification number |
| condom with plastic bag | - | - |
| tattoo sleeve | biometric data | biometric data |
| transcript | student transcript | address, date of birth, education level, email address, name, school attended |
| business card | employment information | address, email address, employment history, employer name, name, phone number, position |
| condom box | - | - |
| local newspaper | - | - |
| medical record document | medical information, medical prescription, medical record | address, date of birth, diagnosis, email address, name, personal identification number |
| id card | personal identification information | address, date of birth, driver license, driver license number, name, personal identification number, photo, signature, gender |

Table 6: Private Object Categories with Synonyms and Associated PII Types

However, our framework is not perfect and can also be prone to errors(see Figure 8). As previously discussed, some of them are due to blurriness, labeling mistakes, and fine-grained issues such as misplacement or localization. Specifically, Figure 8a-8d presents a blurry document, where it misses a lot of fine-grained details. In Figure 8e-8h, the framework successfully captures a fine-grained mask but the labeling fails in the card of the name resulting in an incomplete High-Risk mask. Finally, some errors are due to limitations of the tools Figure 8j-8p, which produce fine-grained maks misplacement or too coarse masks.

# H  FigPriv and Non-Agentic Approaches

We identify significant limitations of current state-of-the-art models in accurately recognizing and localizing information present in challenging images. While it is possible that a non-agentic approach could also prove effective, we show that current non-agentric approaches do not perform well, and that our proposed agentic approach does. Figure 9 shows visual comparison with other methods. In particular, we compare with Gemini2.5 (Comănici & et al., 2025), GPT4o (OpenAI, 2024) and MistralOCR (Mistral AI, 2025) which are known models recently released. Our multi-agent system achieves precise content localization, enabling the context-aware masking required for effective privacy preservation while maximizing content utility, critical for BLV users.

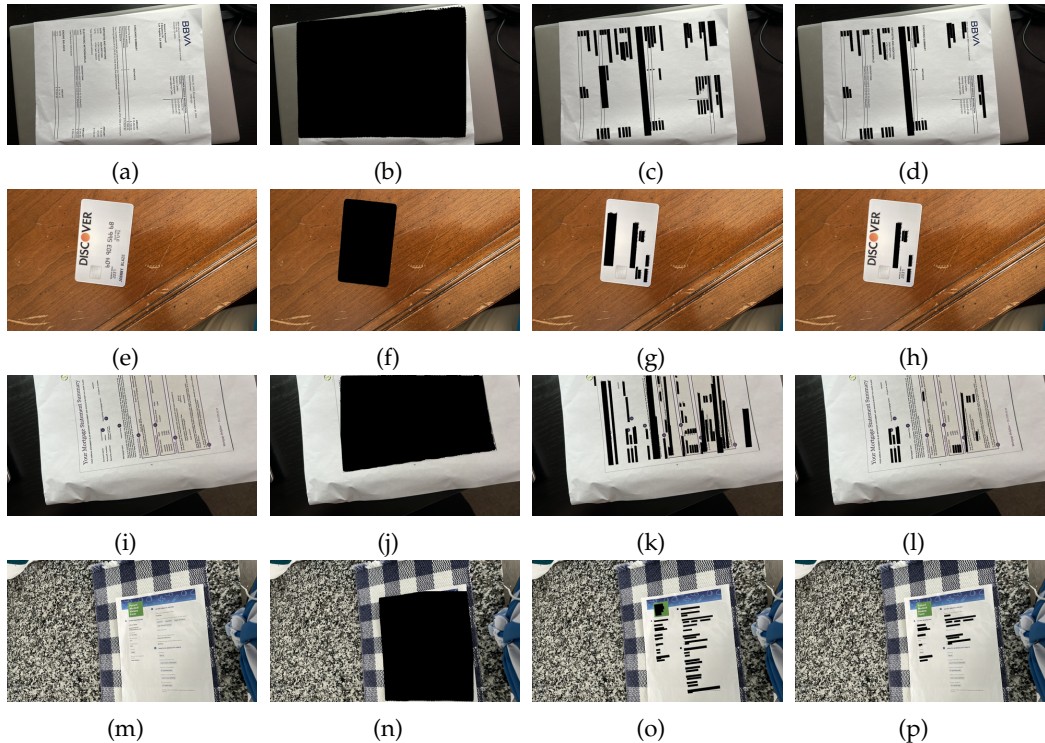

Figure 7: Examples of the strengths of `FiG-Priv`. From left to right: the Full image, the Object Mask within the image, the Fine-grained mask within the image, and the High-Risk mask within the image. In the first row framework manage to produce a fine-grained mask for a large amount of content. In second row it succesfully discerns high-risk masks from fine-grained ones. In the third and forth row, it performs both tasks effectively, even on slightly rotated documents.

## I    Computational Overhead of `FigPriv`

We randomly sample 100 images from the BIVPrigSeg (Tseng et al., 2025) dataset and run our pipeline on each, recording the processing time per image. On average, the pipeline takes 77.51 seconds per image. The fastest sample is processed in 11.19 seconds, while the slowest takes 616.53 seconds. Figure 10 shows a histogram of the processing times. The distribution is right-skewed, favoring shorter times, with a standard deviation of 90.71 seconds. Notably, images with longer processing times typically contain a high density of fine-grained information, such as detailed financial reports. While our framework is not currently suitable for real-time applications, our goal is to show the feasibility of such fine-grained privacy framework.

## J    Annotation Questions and Rating Scales

To answer **RQ-PrivProt**, we manually annotate images processed by three different masking strategies with questions and rating scales as shown in Table 7. Note that when fine-grained masking is not necessary ("No" for question 2) the rest of the questions are skipped.

## K    Visual Question Answering Templates

To correctly assess the preservation benefit of non-private information, we rely on the confidence of VLMs under the different masking approaches. We design several questions based on private categories. Each of these questions focuses on specific aspects of the

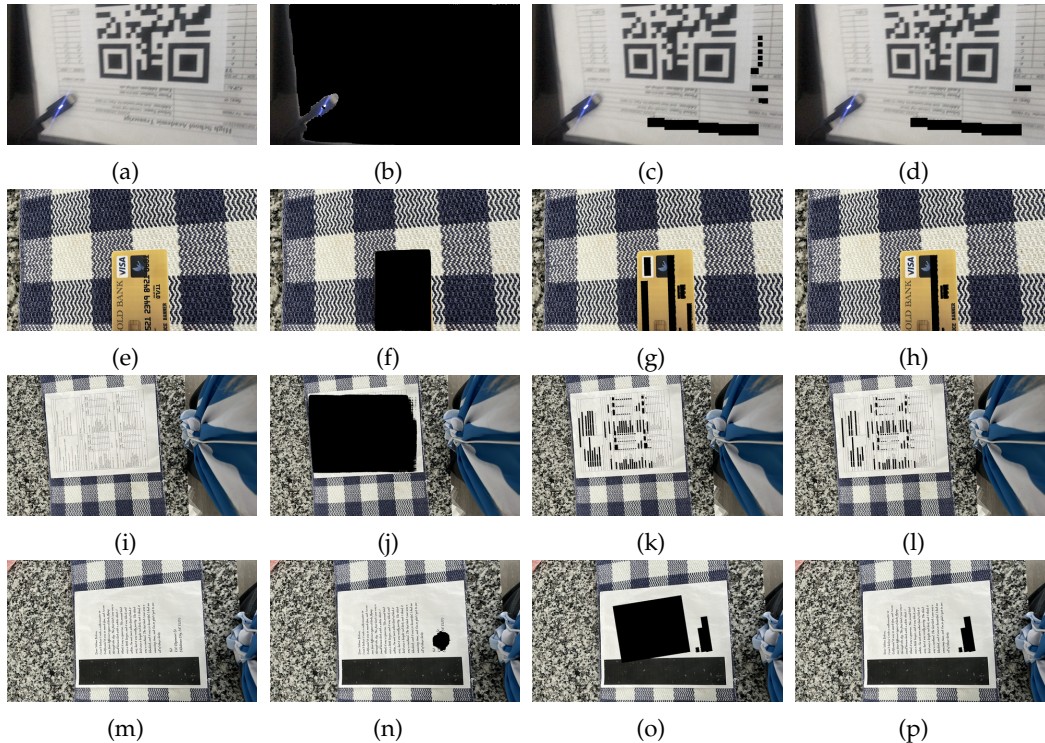

|     |     |     |     |
| --- | --- | --- | --- |
| (a) | (b) | (c) | (d) |
| (e) | (f) | (g) | (h) |
| (i) | (j) | (k) | (l) |
| (m) | (n) | (o) | (p) |

Figure 8: Examples of the limitations of FiG-Priv. From left to right: the Full image, the Object Mask within the image, the Fine-grained mask within the image, and the High-Risk mask within the image. The issue in the first row was caused by blurriness. The second row issue resulted from mislabeling. The third row issue arose from mask misplacement. The fourth row issue was due to mislocalization.

| Annotation Questions | Rating Scale |
| --- | --- |
| Is full masking correct? | 1–Completely off the masking; 2–Poor masking; 3–partially masking; 4-Mostly proper masking; 5–Perfect masking |
| Is fine-grained masking necessary? | 0-No; 1-Yes |
| Is fine-grained masking correct? | 1–Completely off the masking; 2–Poor masking; 3–partially masking; 4-Mostly proper masking; 5–Perfect masking |
| Is high-risk masking correct? | 1–Completely off the masking; 2–Poor masking; 3–partially masking; 4-Mostly proper masking; 5–Perfect masking |

Table 7: Annotation Questions and Rating Scales

category, which usually do not overlap with other ones. E.g., questions about the card number on a credit card or the kind of medications in doctor prescriptions. The full list of questions along with the QwenVL2.5 7B confidence are shown in Table 8-19.

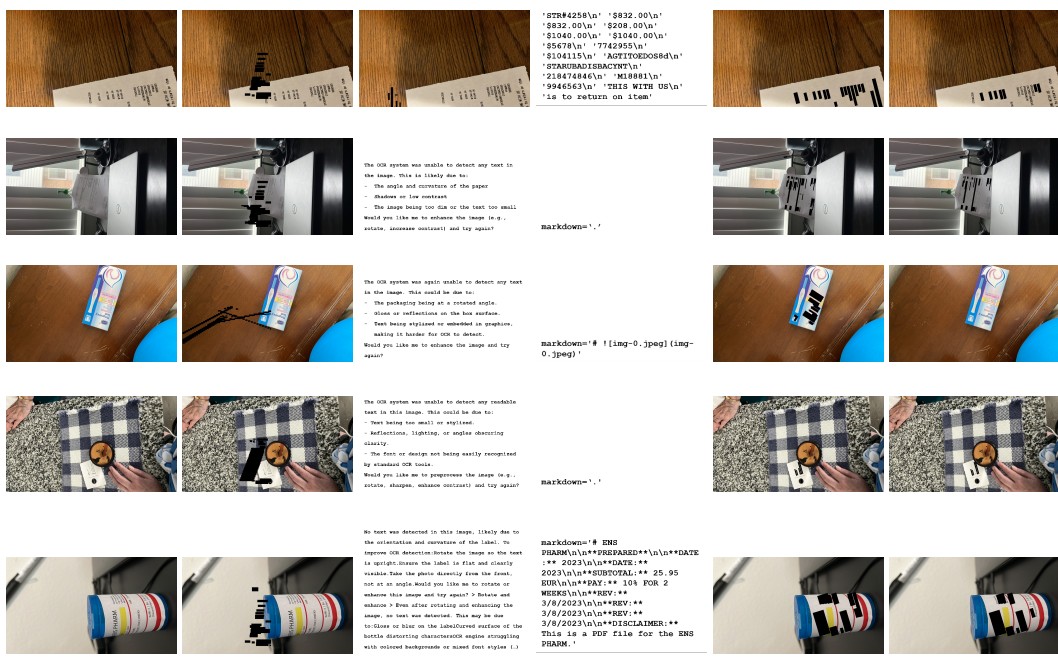

Figure 9: Comparison results with other methods. From left to right: the Full image, Gemini 2.5 output, GPT-4o output, MistralOCR output, Full Fine-Grained output and `FigPriv`. Our two methods visually outperforms the other ones.

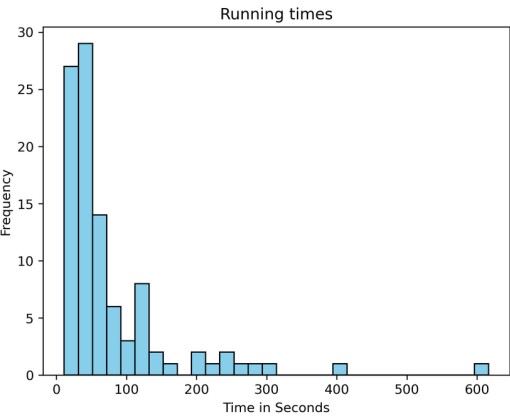

Figure 10: Running times: Time vs Frequency

| Question | Full Image | Object Mask | Fine-grained Mask | High-Risk Mask |
|---|---|---|---|---|
| Given this question: 'What does this say?', based on the image, are you able to answer the question? | 0.2150 | 0.0201 | 0.0388 | 0.1399 |
| Given this question: 'I need to identify the kind of form I have here. Can you provide me with just a few words that you can read on this piece of paper?', based on the image, are you able to answer the question? | 0.6327 | 0.0563 | 0.2107 | 0.5699 |
| Given this question: 'Who is this from?', based on the image, are you able to answer the question? | 0.0903 | 0.0031 | 0.0068 | 0.0459 |
| Given this question: 'What are the amounts?', based on the image, are you able to answer the question? | 0.3747 | 0.0114 | 0.1021 | 0.1089 |
| Given this question: 'Read the statement.', based on the image, are you able to answer the question? | 0.3166 | 0.0600 | 0.0768 | 0.2197 |

Table 8: Answerable questions on images labeled as *Bank Statement*

| Question | Full Image | Object Mask | Fine-grained Mask | High-Risk Mask |
|---|---|---|---|---|
| Given this question: 'What does this say?', based on the image, are you able to answer the question? | 0.1351 | 0.0861 | 0.0219 | 0.0899 |
| Given this question: 'Is this piece of mail?', based on the image, are you able to answer the question? | 0.4130 | 0.1936 | 0.1550 | 0.2954 |
| Given this question: 'Can you tell me who this is from?', based on the image, are you able to answer the question? | 0.1605 | 0.0963 | 0.0244 | 0.0177 |
| Given this question: 'I just wanted to know who the return address is on the, or what the return address is on this envelope on the upper left hand corner.', based on the image, are you able to answer the question? | 0.0971 | 0.0446 | 0.0167 | 0.0123 |
| Given this question: 'What is this mail, where is it from?', based on the image, are you able to answer the question? | 0.1735 | 0.0792 | 0.0309 | 0.0565 |
| Given this question: 'Who is this mail for?', based on the image, are you able to answer the question? | 0.1618 | 0.0480 | 0.0152 | 0.0657 |

Table 9: Answerable questions on images labeled as *Letter with Address*

| Question | Full Image | Object Mask | Fine-grained Mask | High-Risk Mask |
|---|---|---|---|---|
| Given this question: 'What is this?', based on the image, are you able to answer the question? | 0.7965 | 0.7884 | 0.7695 | 0.7998 |
| Given this question: 'What kind of card is this', based on the image, are you able to answer the question? | 0.7472 | 0.0017 | 0.1342 | 0.6498 |
| Given this question: 'What is the expiration date?', based on the image, are you able to answer the question? | 0.3396 | 0.0001 | 0.0139 | 0.0299 |
| Given this question: 'Is there a phone number on this card and what is it?', based on the image, are you able to answer the question? | 0.0423 | 0.0001 | 0.0083 | 0.0082 |
| Given this question: 'Can you please tell me the 1-800 number on this card?', based on the image, are you able to answer the question? | 0.0871 | 0.0001 | 0.0021 | 0.0006 |
| Given this question: 'Can you read this card number?', based on the image, are you able to answer the question? | 0.0855 | 0.0002 | 0.0005 | 0.0004 |

Table 10: Answerable questions on images labeled as *Credit or Debit Card*

| Question | Full Image | Object Mask | Fine-grained Mask | High-Risk Mask |
|---|---|---|---|---|
| Given this question: 'What does this say?', based on the image, are you able to answer the question? | 0.3337 | 0.0243 | 0.0672 | 0.1405 |
| Given this question: 'What bill is this?', based on the image, are you able to answer the question? | 0.1673 | 0.0004 | 0.0105 | 0.0514 |
| Given this question: 'How much is this bill for?', based on the image, are you able to answer the question? | 0.4302 | 0.0002 | 0.0572 | 0.1054 |
| Given this question: 'What is the total amount?', based on the image, are you able to answer the question? | 0.4168 | 0.0014 | 0.0660 | 0.1179 |
| Given this question: 'I know that this is a receipt, but what is it a receipt of?', based on the image, are you able to answer the question? | 0.1733 | 0.0011 | 0.0271 | 0.0843 |

Table 11: Answerable questions on images labeled as *Bills or Receipt*

| Question | Full Image | Object Mask | Fine-grained Mask | High-Risk Mask |
|---|---|---|---|---|
| Given this question: 'What does this say?', based on the image, are you able to answer the question? | 0.3196 | 0.0182 | 0.0644 | 0.3161 |
| Given this question: 'What is the expiration date?', based on the image, are you able to answer the question? | 0.0011 | 0.0030 | 0.0031 | 0.0013 |

Table 12: Answerable questions on images labeled as *Pregnancy Test Box*

| Question | Full Image | Object Mask | Fine-grained Mask | High-Risk Mask |
|---|---|---|---|---|
| Given this question: 'What does this say?', based on the image, are you able to answer the question? | 0.2754 | 0.0584 | 0.0520 | 0.2051 |
| Given this question: 'I need to identify the kind of form I have here. Can you provide me with just a few words that you can read on this piece of paper?', based on the image, are you able to answer the question? | 0.6921 | 0.1721 | 0.2616 | 0.6462 |
| Given this question: 'Who is this from?', based on the image, are you able to answer the question? | 0.0755 | 0.0116 | 0.0146 | 0.0283 |

Table 13: Answerable questions on images labeled as *Mortage or Investment Report*

| Question | Full Image | Object Mask | Fine-grained Mask | High-Risk Mask |
|---|---|---|---|---|
| Given this question: 'What does this say?', based on the image, are you able to answer the question? | 0.3403 | 0.0831 | 0.0335 | 0.1040 |
| Given this question: 'Can you read the name of this prescription?', based on the image, are you able to answer the question? | 0.1456 | 0.0100 | 0.0003 | 0.0007 |
| Given this question: 'What kind of medications?', based on the image, are you able to answer the question? | 0.0889 | 0.0191 | 0.0019 | 0.0052 |
| Given this question: 'What are the instructions?', based on the image, are you able to answer the question? | 0.2047 | 0.0661 | 0.0776 | 0.0490 |
| Given this question: 'Whatś the side effects?', based on the image, are you able to answer the question? | 0.0011 | 0.0032 | 0.0011 | 0.0014 |

Table 14: Answerable questions on images labeled as *Doctor Prescription*

| Question | Full Image | Object Mask | Fine-grained Mask | High-Risk Mask |
|---|---|---|---|---|
| Given this question: 'What does this say?', based on the image, are you able to answer the question? | 0.0698 | 0.0267 | 0.0106 | 0.0511 |
| Given this question: 'Can you read what kind of medicine is in the bottle? If you can only tell me what letter it starts with Iĺl know the rest.', based on the image, are you able to answer the question? | 0.0782 | 0.0279 | 0.0181 | 0.0310 |
| Given this question: 'What kind of medication is it?', based on the image, are you able to answer the question? | 0.0116 | 0.0056 | 0.0001 | 0.0001 |
| Given this question: 'What are the instructions on this bottle?', based on the image, are you able to answer the question? | 0.0346 | 0.0165 | 0.0023 | 0.0026 |
| Given this question: 'Whatś the side effects on this medicine bottle?', based on the image, are you able to answer the question? | 0.0006 | 0.0003 | 0.0005 | 0.0005 |

Table 15: Answerable questions on images labeled as *Empty Pill Bottle*

| Question | Full Image | Object Mask | Fine-grained Mask | High-Risk Mask |
|---|---|---|---|---|
| Given this question: 'What does this say?', based on the image, are you able to answer the question? | 0.2080 | 0.0661 | 0.0769 | 0.1213 |

Table 16: Answerable questions on images labeled as *Transcript*

| Question | Full Image | Object Mask | Fine-grained Mask | High-Risk Mask |
|---|---|---|---|---|
| Given this question: 'What does this say?', based on the image, are you able to answer the question? | 0.3636 | 0.0078 | 0.0118 | 0.0257 |
| Given this question: 'What does this business card have on it?', based on the image, are you able to answer the question? | 0.6101 | 0.0081 | 0.1436 | 0.2021 |
| Given this question: 'What is the card in my hand?', based on the image, are you able to answer the question? | 0.0872 | 0.0014 | 0.0050 | 0.0087 |
| Given this question: 'What is the phone number on this business card? Can you please tell me who this business card is from? What does this say? Iḿ looking for a phone number and email address. Thank you.', based on the image, are you able to answer the question? | 0.4949 | 0.0052 | 0.0159 | 0.0490 |

Table 17: Answerable questions on images labeled as *Business Card*

| Question | Full Image | Object Mask | Fine-grained Mask | High-Risk Mask |
|---|---|---|---|---|
| Given this question: 'What does this say?', based on the image, are you able to answer the question? | 0.1943 | 0.0492 | 0.0716 | 0.1895 |
| Given this question: 'Can you please describe whatś on this newspaper?', based on the image, are you able to answer the question? | 0.4415 | 0.1346 | 0.2809 | 0.4037 |
| Given this question: 'Whatś the date of this paper?', based on the image, are you able to answer the question? | 0.0815 | 0.0253 | 0.0581 | 0.0693 |
| Given this question: 'What is this newspaper section about?', based on the image, are you able to answer the question? | 0.2450 | 0.0526 | 0.0931 | 0.2099 |
| Given this question: 'What newspaper is this?', based on the image, are you able to answer the question? | 0.1088 | 0.0225 | 0.0484 | 0.1007 |

Table 18: Answerable questions on images labeled as *Local Newspaper*

| Question | Full Image | Object Mask | Fine-grained Mask | High-Risk Mask |
|---|---|---|---|---|
| Given this question: 'What does this say?', based on the image, are you able to answer the question? | 0.3017 | 0.0215 | 0.0477 | 0.1984 |
| Given this question: 'I need to identify the kind of form I have here. Can you provide me with just a few words that you can read on this piece of paper?', based on the image, are you able to answer the question? | 0.7348 | 0.0876 | 0.2712 | 0.6707 |
| Given this question: 'Who is this from?', based on the image, are you able to answer the question? | 0.0958 | 0.0067 | 0.0082 | 0.0106 |

Table 19: Answerable questions on images labeled as *Medical Record Document*

