# OpenReview forum: "Beyond Blanket Masking: Examining Granularity for Privacy Protection in Images Captured by Blind and Low Vision Users"
_colmweb.org/COLM/2025/Conference — COLM 2025_

### Official Review · Reviewer_MxCN · 2025-05-07

**Rating:** 6
**Confidence:** 4
**Ethics Flag:** 1

**Summary:**

This work introduces FiG-Priv a new method to mask potentially private information contained in images (particularly focusing on VLM usage of vision-impaired users). It makes two primary contributions in this regard: (1) A new taxonomy that identifies the severeness and relationship of various PII types (based on page rank on the ITAP dataset). Further, they (2) present an agentic framework that uses multiple local LLMs alongside OCR tools to identify, classify, and ultimately mask parts of the image considered high-risk. Their evaluation on a newly adapted dataset shows that high-risk classification outperforms baselines, keeping higher utility (on non-PII related questions) while retaining comparable privacy – a result that is confirmed by a human study.

**Questions To Authors:**

Any of the points raised in the section Weaknesses can be phrased as a corresponding question. Further:
- It would be interesting to see whether we could receive overall better results if, instead of just masking, we would actually infill realistic data such that later inference models are actually less "out of distribution."
- How well do the authors believe this approach could be extended beyond just high-risk PII directly contained in the image but rather image parts that more implicitly leak personal data? In a similar vein, it would be interesting to see how the work performs in less BLV-focused settings.

**Reasons To Accept:**

- The issue at hand is highly relevant. VLM are increasingly used by users who knowingly and unknowingly expose their sensitive information to them. Preserving individuals' privacy while maintaining as much of the utility as possible in these images is a valuable effort.
- The proposed method clearly reduces the amount of information anonymized (higher utility) while maintaining privacy for high-risk PII types.
- The inclusion of a human study is especially relevant when dealing with utility-focused anonymizations.

**Reasons To Reject:**

- While generally appreciated, the current evaluation of utility feels suboptimal:
	- The presented questions are directly aimed at aspects that are not considered private and ultimately asked/evaluated only in a binary format (sensible for the "is a private object contained" setting – less sensible for the "could this be answered" setting where such an evaluation can introduce a lot of noise compared to classical ground-truth evals).
	- Also the final dataset used here is not very large (168 samples).
	- Further most of the questions cannot even be answered by the model with access to the full image (questioning if we should be asking these questions in the first place)
- While the authors compare against full and partial masking, various natural baselines are missing (e.g., stronger VLM zero shotting the masks, various OCR tools, iterative adversarial inference -> anonymization)
- Given the models used in the evaluation it seems like the masking model is significantly larger (Qwen2.5 VL 72B) than any of the models used for later inferences. This raises the question why this approach is practical, when users would have to apply such a strong local baseline model in the first place (and could just use it directly to answer their query). It would be sensible to observe the performance of the approach in settings where the models performing anonymization are actually smaller than what is used for later detection
- Similarly, the work includes no discussion of the costs of actually running the framework (w.r.t. inference-cost and time)

---

> ### Author Response · Authors · 2025-06-03
>
> We thank Reviewer MxCN for all the provided feedback. We address the concerns and questions below.
>
> **Evaluation of utility feels suboptimal**: Thank you for pointing this out. Given that we don’t have ground-truth questions and answers for the images, we measure whether the image content and text are **semantically aligned**, along with the **inter-annotator agreement** for the human annotations.
>
> The evaluation results are obtained by computing the likelihood of the models to answer binary questions, which can be regarded as *confidence scores*, given that *the output is a probability of correctness*. We also include synthetic objects that are easily recognizable as a control set; the likelihood of recognizing the control object under varying masking conditions serves as an indicator of the model’s visual/language alignment. Table 3 shows that the models are more confident in identifying the control object when the private information is completely masked, lowering their confidence as the fine-grained and selective-fine-grained masking is applied.
>
> The final dataset corresponds to **the whole Biv-Priv-Seg dataset (both support and query sets = 1072 images)**. The 168 samples correspond to the human-evaluated subset.
> _____________
> **Most of the questions cannot even be answered by the model with access to the full image**: we show in Table 3 that models are capable of answering both existence and utility with high confidence. Both Qwen2.5-VL and Gemma-3 yield a very high likelihood, with *80.6%* and *91.5%*, respectively. We include LlaVA-1.6, which struggles the most with 57.2%, given the popularity of this model in prior literature, and find that, in a model with less parameters and trained with less data, full-masking hurts the most (dropping to 3.5%), while our selective high-risk mask dramatically improves the model performance (to a 43.7%) without PII exposure. This further justifies our fine-grained and selective masking exploration.
> _______________
> **Various natural baselines are missing**: We kindly ask the reviewer to help us identify the state-of-the-art methods able to perform selective fine-grained masking in images taken by BLV users, or any image taken under challenging conditions. It is difficult to tell from the brief parenthetical exactly what is intended, though some of these sound like separate research projects in their own right.
>
> To the best of our knowledge, there is no prior work that addresses this issue. As mentioned, we run several experiments using proprietary models and paywall APIs, results available in this link: **https://figpriv.github.io**.
>
> - *MistralOCR*, the SOTA model specialized for OCR, is unable to output bounding boxes, but is able to read some text. This model’s ability to parse content in challenging images emphasizes the need for better fine-grained de-identification of sensitive content.
> - *Gemini 2.5 Flash Preview 05-20*, the latest SOTA model by Google, is able to identify the text and output bounding boxes, but the resulting bboxes are completely unaligned, or result on blanket-masking. Without precise bboxes, selective masking is impossible.
> - *ChatGPT versions 4o and o3* (including advanced reasoning for up to 10 minutes), are unable to identify most of the content in the given images and fail to output the corresponding bboxes, making selective masking impossible.
>
> These results underscore the importance of providing better systems that enable BLV users' privacy and utility in the way they interact with the world. Our work and proposed framework is the first attempt to address this.
>
> Furthermore, *iterative adversarial inference for content anonymization* [1] has shown promising results in ideal scenarios, where text is perfectly visible and both VLMs and OCR systems are able to detect all content. This is significantly different from our current scenarios, with images taken under challenging conditions, including occlusions, images taken with low light, partially observable, blurred, etc. Furthermore, our framework keeps all original content in the image, and is able to localize, transform, mask and recover the image content for semantic recognition; Iterative adversarial inference only rewrites the content, which reduces the real utility of the image (incorporating false information [2]), and is naturally unable to localize and recover the original semantic content provided [3].
>
> [1] Li, Xiaoting, Lingwei Chen, and Dinghao Wu. "Adversary for social good: Leveraging adversarial attacks to protect personal attribute privacy." Acm Transactions on Knowledge Discovery from Data 18.2 (2023): 1-24.
>
> [2] Frikha, Ahmed, et al. "IncogniText: Privacy-enhancing Conditional Text Anonymization via LLM-based Private Attribute Randomization." arXiv preprint arXiv:2407.02956 (2024). - NeurIPS 2024.
>
> [3] Staab, Robin, et al. "Large language models are advanced anonymizers." arXiv preprint arXiv:2402.13846 (2024). - ICLR 2025.

---

> > ### Author Response · Authors · 2025-06-03
> >
> > **Why this approach is practical**: We agree that in practice, such system requires small models, but our goal is to show the feasibility of such fine-grained privacy framework; our goal is to lay the groundwork and encourage subsequent research to continue refining and deploying these methods with smaller architectures.
> >
> > While smaller models are unable to localize fine-grained information, large VLMs are able to detect objects, and specialized models like MistralOCR can parse the content in challenging images. As models grow bigger, their ability and possible misuse raise concerns.
> >
> > To the best of our knowledge, this is the first study that accounts for privacy and utility by examining and exploring how fine-grained identification and subsequent masking enable BLV users to leverage AI systems without exposing PII, while maintaining privacy and reducing financial harm exposure.
> >
> > Our exploration is intended to account for both large SOTA VLMs, specialized OCR, and a multi-agent system to enable fine-grained selective masking under challenging conditions. Thus, our exploration also includes smaller VLMs to further assess the utility of the resulting images on accessible models, which are more likely to run on consumer-based systems.
> > _______
> > **Cost**:Fig-Priv runs on a server with 4 A100 GPUs, with most samples requiring up to ~50 seconds for full inference. By comparison, GPTo3 can take up to ~7 minutes for a single sample, producing errors on cases Fig-Priv correctly processes in ~40 seconds. We will include a full comparison of cost and inference times with leading SOTA models, such as Chain-of-Thought (e.g., Gemini 2.5, GPTo3) and multi-tooling APIs (e.g., GPT-4o), in our final version.
> > _______
> > **Infill realistic data instead of masking**: We agree that it would be an interesting follow-up work to test model performance under various forms of obfuscation, such as removing or infilling high-risk regions of an image. In fact, prior work has explored this through user studies. In [4], the authors perform coarse-grained masking, removing all the visible object, and found that this approach leads the models to produce inconsistent descriptions of objects, adding confusion to the BLV user. This is part of our motivation for fine-grained detection and masking, as our selective masking keeps most of the image content intact.
> >
> > Orthogonally, prior work [5] explored infilling the sensitive content in a given image via generative models, but it does not account for faithfulness, user perception, or utility, as the generated content can hallucinate undesirable content. Other works have proposed infilling images that portray humans [6, 7], or even completely removing people from images [8]. While those works are biometric-centric, we support fine-grained object localization in BLV images, featuring real-world objects, in photos taken by BLV users, which include occluded, blurred, and out-of-focus content.
> >
> > We agree that this is a direction that deserves further exploration, and we will include this in our discussion on opportunities and open challenges in the final version.
> >
> > [4] Zhang, Lotus, Abigale Stangl, Tanusree Sharma, Yu-Yun Tseng, Inan Xu, Danna Gurari, Yang Wang, and Leah Findlater. 2024. “Designing Accessible Obfuscation Support for Blind Individuals’ Visual Privacy Management.” Pp. 1–19 in Proceedings of the CHI Conference on Human Factors in Computing Systems. Honolulu HI USA: ACM.
> >
> > [5] Yang, Guang, et al. "Eraser: adversarial sensitive element remover for image privacy preservation." Proceedings of the AAAI Conference on Artificial Intelligence. Vol. 37. No. 12. 2023.
> >
> > [6] Sun, Qianru et al. “Natural and Effective Obfuscation by Head Inpainting.” 2018 IEEE/CVF Conference on Computer Vision and Pattern Recognition (2017): 5050-5059.
> >
> > [7] Rot, Peter et al. “PrivacyProber: Assessment and Detection of Soft–Biometric Privacy–Enhancing Techniques.” IEEE Transactions on Dependable and Secure Computing 21 (2022): 2869-2887.
> >
> > [8] Ertan, Murat Bilgehan, et al. "Beyond Anonymization: Object Scrubbing for Privacy-Preserving 2D and 3D Vision Tasks." arXiv preprint arXiv:2504.16557 (2025).
> > ____
> >
> > **Extend beyond just high-risk PII but rather image parts that more implicitly leak personal data**: Thank you for pointing this out! Yes, we mention this in our paper (L367-373). It is feasible to include non-textual high-risk information by personalizing the high-risk score metadata, so that the Detection and Segmentation module accounts for this information and apply a full mask to those specific objects. We will make all our code available, including instructions for adding personalized masking criteria. We will further expand on how to incorporate this in our final version.

---

> > ### Comment · Reviewer_MxCN · 2025-06-06
> > **Thank you for your answers**
> >
> > I thank the authors for the comprehensive answer and will try to respond in the order of points raised:
> >
> > - Utility: I agree, and I appreciate the effort made - my point here was primarily about the contextualization of RQ-Perf Figure 5. Here, proper filtering of questions to answerable (ground truth) would have helped (at least my) understanding significantly.
> > - Various natural baselines missing: I think my point might not have been entirely clear here. I did not want to say, "There exists a range of works that already does this; please compare to these (baselines)," I rather meant to say "there are many system-wise simpler approaches (that may or may not work), you should demonstrate that your approach brings advantages to what people might naively try." Independently, I think your additional experiments exactly demonstrate what I wanted to see, i.e., that there is an actual advantage over these more naive approaches. This also solidifies the original contribution of the work. I would love to have extended experimental numbers on these in an updated version of the work, thus establishing a full-on comparison of methods.
> >
> > Conclusion: Especially based on the new examples, I think this is valuable work, and I will accordingly raise my score to favor acceptance. My concerns with respect to runtime / practicality partially remain, but given this is an early exploration into this direction, I believe this is alright.

---

> > > ### Author Response · Authors · 2025-06-08
> > >
> > > Thank you for your response and taking the time to provide your final feedback! We are grateful for your decision to raise the score and for your positive assessment. We will gladly incorporate your final suggestions into the camera-ready version.

---

### Official Review · Reviewer_BRNH · 2025-05-11

**Rating:** 6
**Confidence:** 3
**Ethics Flag:** 1

**Summary:**

This paper presents a privacy protection framework, FiG-Priv, to protect images captured by blind and low-vision (BLV) users using VLM-based visual assistant systems. It stands out from prior works by providing fine-grained identification and masking of high-risk personally identifiable information (PII), while baselines mainly apply coarse masking. It preserves more visible content, enabling the VLM to provide more useful responses. The approach involves a sequence of agents that detect, segment, and process private objects from captured images, as well as a data-driven risk scoring mechanism to quantify PII risk and selectively mask the most sensitive content. Evaluations demonstrate that FiG-Priv increases the answerability of VLMs and maintains comparable privacy protection to full masking.

**Questions To Authors:**

The proposed method mainly focuses on textual information. How can non-textual high-risk information (e.g., facial appearance) be identified and obfuscated? Incorporating techniques to protect such information would significantly improve the paper’s generalizability and social impact.

**Reasons To Accept:**

- This paper addresses privacy issues in assistive technologies, which is important, insightful, and socially impactful.
- The technical design is sound. Though the proposed framework is largely based on existing techniques, the authors organize them effectively and achieve good results.
- The evaluations show promising outcomes in both privacy protection and model usability.

**Reasons To Reject:**

- The paper’s technical contribution is quite incremental, as it mostly combines existing techniques (e.g., OCR, segmentation, Identity Ecosystem Graph, PageRank).
- The authors compared FiG-Priv to none of the state-of-the-art methods, but merely two baselines.
- Though the results are promising, the evaluations rely heavily on the BIV-Priv-Seg dataset, which appears to be small-scale and may not fully reflect real-world scenarios. For example, the reviewer examined the dataset and found that the images are generally clear and well-lit, which is insufficient to validate the framework’s robustness in more challenging, realistic conditions.
- The reviewer is also concerned about potential statistical bias: the human annotation study appears subjective and lacks clear reliability statistics. Additionally, the risk model and annotated data focus primarily on financial harm (e.g., bank account balance, bill paid, credit card number in Tab. 1), while emotional or subjective user concerns are only briefly acknowledged in the discussion.

---

> ### Author Response · Authors · 2025-06-03
>
> We thank Reviewer BRNH for all the provided feedback. We address the concerns and questions below.
>
> **Contribution**: Our proposed framework covers multiple subject areas focused on language modeling for the second iteration of COLM, including:
>
> **17. LMs with tools and code: integration with tools and APIs**, LM-driven software engineering
>
> **18. LMs on diverse domains and novel applications: visual LMs**, code LMs, math LMs, and so forth, with extra encouragements for less studied domains or applications such as chemistry, medicine, education, database and beyond.
>
> Furthermore, we identify significant limitations of current SOTA models in accurately recognizing and localizing information present in challenging images; **please see SOTA examples here: https://figpriv.github.io/** (*includes examples of paywall APIs and proprietary closed-source models*).
>
> Our multi-agent system achieves precise content localization, enabling the context-aware masking required for effective privacy preservation while maximizing content utility, critical for BLV users.
> ___________
>
> **No SOTA comparisons**: We kindly ask the reviewer to help us identify the state-of-the-art methods able to perform selective fine-grained masking in images taken by BLV users, or any image taken under challenging conditions.
>
> To the best of our knowledge, there is no prior work that addresses this issue. As mentioned, we run several experiments using proprietary models and paywall APIs, results available in this link: **https://figpriv.github.io**.
>
> - *MistralOCR*, the SOTA model specialized for OCR, is unable to output bounding boxes, but is able to read some text. This model’s ability to parse content in challenging images emphasizes the need for better fine-grained de-identification of sensitive content.
> - *Gemini 2.5 Flash Preview 05-20*, the latest SOTA model by Google, is able to identify the text and output bounding boxes, but the resulting bboxes are completely unaligned, or result on blanket-masking. Without precise bboxes, selective masking is impossible.
> - *ChatGPT versions 4o and o3* (including advanced reasoning for up to 10 minutes), are unable to identify most of the content in the given images and fail to output the corresponding bboxes, making selective masking impossible.
>
> These results underscore the importance of providing better systems that enable BLV users' privacy and utility in the way they interact with the world. Our work and proposed framework is the first attempt to address this.
> __________
> **Is the BIV-Priv-Seg dataset enough?**: To the best of our knowledge, the BIV-Priv-Seg dataset is the only real-world dataset that comprises images taken by BLV users and contains PII information. The objects in these images are often *occluded, zoomed in with content cropped out, blurry, in low lighting conditions, misaligned, among other challenges* (https://figpriv.github.io/bivpriv_samples.png). Moreover, all other splits from the VizWiz dataset have already been masked and PII removed.
>
> Other existing datasets collected by BLV users are designed for outdoor navigation [1], scene understanding [2], or object recognition [3]; none of these datasets contains PII. We kindly ask the reviewer to help us identify other datasets with this type of information.
>
> [1] Islam, Md Touhidul, et al. "A Dataset for Crucial Object Recognition in Blind and Low-Vision Individuals' Navigation." arXiv preprint arXiv:2407.16777. ASSETS ‘24.
>
> [2] Ruei-Che Chang, Yuxuan Liu, and Anhong Guo. 2024. WorldScribe: Towards Context-Aware Live Visual Descriptions. In Proceedings of the 37th Annual ACM Symposium on User Interface Software and Technology. UIST '24.
>
> [3] Theodorou, L., Massiceti, D., Zintgraf, L. , Stumpf, S. , Morrison, C., Cutrell, E., Harris, M. T. & Hofmann, K. (2021). Disability-first Dataset Creation: Lessons from Constructing a Dataset for Teachable Object Recognition with Blind and Low Vision Data Collectors. In: The 23rd International ACM SIGACCESS Conference on Computers and Accessibility. ASSETS '21.
> ________

---

> > ### Author Response · Authors · 2025-06-03
> >
> > **Human annotations and statistical bias**: Thank you for pointing this out! Yes, as we mentioned in our paper (footnote on page 1), the information collected and provided by the Identity Ecosystem Graph only accounts for financial loss, since emotional distress is subjective and difficult to measure.
> >
> > Our human annotations follow the standard statistical protocol [4, 5, 6] and measure inter-annotator agreement for the information that we can evaluate: bbox precision, mask coverage, readability, and utility (Appendix G, page 19 in our paper). Furthermore, 74.5% of the private objects in the Biv-Priv-Seg dataset are text-related, which emphasizes the importance of properly managing this type of information. We will add more clarifications in the final version.
> >
> > [4] Carletta, J. 1996. “Assessing Agreement on Classification Tasks: The Kappa Statistic.” Computational Linguistics 22(2): 249-254.
> >
> > [5] McHugh ML. Interrater reliability: the kappa statistic. Biochem Med (Zagreb). 2012;22(3):276-82. PMID: 23092060; PMCID: PMC3900052.
> >
> > [6] R. Bernardi, R. Cakici, D. Elliott, A. Erdem, E. Erdem, N. Ikizler-Cinbis, F. Keller, A. Muscat, and B. Plank, “Automatic description generation from images: A survey of models, datasets, and evaluation measures,” Journal of Artificial Intelligence Research, vol. 55, pp. 409–442, 2016.
> > _______
> > **Non-textual high-risk information**: Thank you for pointing this out! Yes, we mention this in our paper (L367-373). It is feasible to include non-textual high-risk information by personalizing the high-risk score metadata, so that the Detection and Segmentation module accounts for this information and apply a full mask to those specific objects. We will make all our code available, including instructions for adding personalized masking criteria. We will further expand on how to incorporate this in our final version.

---

> ### Author Response · Authors · 2025-06-09
>
> Dear Reviewer BRNH,
>
> We want to thank you for your valuable comments and feedback on our submission. As the rebuttal period is coming to an end, we wanted to follow up to see if you have any additional questions or concerns about our responses. If there’s anything further you’d like us to clarify, we’d be happy to address it. Thank you again for your time and for reviewing our paper.

---

> > ### Comment · Reviewer_BRNH · 2025-06-10
> > **Thanks for your patient response!**
> >
> > Thanks for your detailed feedback, I would like to raise my original.

---

### Official Review · Reviewer_LRXX · 2025-05-14

**Rating:** 6
**Confidence:** 3
**Ethics Flag:** 1

**Summary:**

The authors identify the increasing amount of blind and low-vision users making use of object detectors.
These technologies often leak private information.
The authors develop a VLM pipeline to redact sensitive or "risky" images from the pipeline.

Instead of a VLM, the pipeline helps extract information and restricts the reading of PII.
The PII that is removed is selectively masked, and only high-risk personally identifiable information (PII) identified by using Pagerank of the types of leaks identified in news articles is removed.
The authors evaluate their framework on the BIV-Priv-Seg dataset and demonstrate that it preserves 26% more image content compared to full-object masking, improves VLM response usefulness by 11%, and enhances the ability to identify image content by 45%, all while maintaining similar privacy protection ratings from human evaluators.

**Questions To Authors:**

Would more fine-grain censorship lead to an increase in privacy?
For example, in the backing domain, is the number of debit payments as much of a privacy leak as the total number?
Is it possible to determine the secondary risk exposure?
To what extent does the ecosystem subgraph capture secondary exposure?
What is the _recall_ for the author's approach?

**Reasons To Accept:**

The authors propose a VLM-driven framework for privacy protection in the context of blind and low-vision users.
Multi-agent systems for user applications are interesting and timely.
The authors compare their pipeline approach to OCR and VLM-based approaches.
Integrating the ecosystem subgraph is an interesting addition that quantifies the risk intuitively.

**Reasons To Reject:**

I question the utility of this entire pipeline and the need for an agentic approach in this context.
The errors in multi-agent systems are challenging to identify, particularly for end users.
The LLMs are so abstract that it is important to communicate error analysis of the classifications, particularly for privacy-focused applications.

For example, Freedom Scientific, which creates the most popular screen reader, has a [feature for identifying objects in images](https://www.freedomscientific.com/picturesmartchallenges/).
There is an explicit privacy tradeoff the BLV users have to make.
The description of technologies on this border should discuss this tradeoff they need to make.

---

> ### Author Response · Authors · 2025-06-03
>
> We thank Reviewer LRXX for all the provided feedback. We address the concerns and questions below.
>
> **The need for an agentic approach in this context**: We identify significant limitations of current SOTA models in accurately recognizing and localizing information present in challenging images. While it is possible that a non-agentic approach could also prove effective, we show that current non-agentric approaches do not perform well, and that our proposed agentic approach does. **Please see SOTA examples here: https://figpriv.github.io/**
>
> Specifically:
> - *Gemini 2.5 Flash Preview (05-20)* consistently produces completely unaligned masking.
> - *GPT-4o* frequently fails to identify text, or masks only a few small, unaligned regions.
> - *GPTo3*, despite long inference times (e.g., 10m to 7m 22s), is often unable to localize the text completely, or results in blanket-masking.
> - *MistralOCR* does not output spatial localization and often struggles with text recognition.
> Our multi-agent system achieves precise content localization, enabling the context-aware masking required for effective privacy preservation while maximizing content utility, critical for BLV users.
>
> ___
>
> **Freedom Scientific and errors in image descriptions**: Thank you for the pointer. *We downloaded the JAWS® Picture Smart AI™ software* provided by Freedom Scientific. We randomly selected 10 images from the BIV-Priv-Seg dataset, and *the software was unable to identify or read the content in those images*. Furthermore, the software does not provide additional information that may help a BLV user to distinguish or identify incorrect output. We understand the challenges with VLMs and hallucinations, and our proposed approach is the first attempt to enable privacy while also ensuring utility. Thus, we provide an in-depth analysis with the given data via manual annotation, object identification (RQ-PrivProt), and VQA performance (RQ-Perf).  *We plan to release our code, all fine-grained scores (output by the VLMs), along with the fine-grained annotations, and corresponding high-risk scores.*
>
> ___
>
> **Would more fine-grained censorship lead to an increase in privacy? + banking domain example**: This is a very interesting question, and we agree that any form of information exposure carries some level of risk. *However,* we base our selective masking on the Identity Ecosystem, which contains the identity attributes most vulnerable to theft, assesses their importance, and determines the personally identifying information (PII) most frequently targeted by thieves (https://identity.utexas.edu/sites/default/files/2020-09/The%20Identity%20Ecosystem.pdf).
> For example, as shown in Figure 6, the combined risk score for credit card information is 100, while the CVV code alone has a much lower score of 29.43. Our use of risk scores in this work is intended to help identify a practical balance between privacy protection and the usability of visual assistant systems.
>
> **Secondary risk exposure**: Short answer: *yes*. To expand on this: in the full Ecosystem Graph (with 1,718 nodes and 18,835 edges), every directed edge (u → v) means “leak u ➞ v later”. Thus, the PageRank influence can flow through arbitrarily long edge chains before it is accumulated at each node. A high PageRank score for “address,” for example, arises precisely because leaking an address propagates to events like stolen mail → bank-account details → fraud (i.e., secondary and tertiary exposures).
>
> **Recall**: Since we don’t have the ground-truth bonding boxes of all fine-grained content (these annotations are not available in the Biv-Priv-Seg dataset), we are not able to compute this number; instead, we conduct a human evaluation to assess privacy protection via Likert scale ratings and high inter-annotator agreement (Table 2, Appendix G).  Furthermore, we evaluate our framework by computing the likelihood of the models to answer binary questions, which can be regarded as confidence scores, given that the output is a probability of correctness, by measuring that the image content and text are semantically aligned.

---

> > ### Comment · Reviewer_LRXX · 2025-06-05
> >
> > Thank you for your replies. Given your response and the replies to others, I will raise my overall score. But let me also address your points.
> >
> > Thank you for sharing the figpriv demo. Looking at these chosen examples help emphasize the need for specialized OCR software within VLM pipelines.
> > With several different components errors could accumulate across the pipeline in unexpected ways. It would be best to address this. (This is what I was inferring in the reasons to reject section.) The Picture Smart AI application was given as a mission critical example of why error analysis is needed on intra pipeline components.
> >
> > Although you are using RageRank and the news articles the authors should acknowledge. the reporting bias (actual vs high-profile leaks) and temporal bias as threat landscape changes. Please also discuss the computational overhead and real-time feasibility of your approach.

---

> > > ### Author Response · Authors · 2025-06-06
> > >
> > > Thank you for your response and engaging in the discussion!
> > >
> > > **Different components and errors**: We are committed to transparency and accountability, critical in frameworks dealing with sensitive information. Thus, we will release all intermediate scores for each component, along with all metadata, all fine-grained scores (output by the VLMs), the fine-grained annotations, and corresponding high-risk scores. We will also make all our code available, including instructions for adding personalized masking criteria and intermediate scores. We will further expand on how to interpret and integrate this in our final version.
> > >
> > > **Reporting bias**: ITAP aggregates data on identity theft from multiple sources (e.g., law enforcement, fraud cases, and news stories). We agree that this is a direction that deserves further exploration, and reporting and temporal bias are important aspects to consider; in this sense, cultural and demographic biases are also interesting aspects that should be accounted for in future work. We will include this in our discussion on opportunities and open challenges in the final version.
> > >
> > > **Computational overhead**: Fig-Priv runs on a server with 4 A100 GPUs, with most samples requiring up to ~50 seconds for full inference. By comparison, GPTo3 can take up to ~7 minutes for a single sample, producing errors on cases Fig-Priv correctly processes in ~40 seconds. We will include a full comparison of cost and inference times with leading SOTA models, such as Chain-of-Thought (e.g., Gemini 2.5, GPTo3) and multi-tooling APIs (e.g., GPT-4o), in our final version.
> > >
> > > **Real-time feasibility**: We agree that in practice, such system requires small models, but our goal is to show the feasibility of such fine-grained privacy framework; our goal is to lay the groundwork and encourage subsequent research to continue refining and deploying these methods with smaller architectures.
> > > While smaller models are unable to localize fine-grained information, large VLMs are able to detect objects, and specialized models like MistralOCR can parse the content in challenging images. As models grow bigger, their ability and possible misuse raise concerns.
> > >
> > > Our exploration is intended to account for both large SOTA VLMs, specialized OCR, and a multi-agent system to enable fine-grained selective masking under challenging conditions. Thus, our exploration also includes smaller VLMs to further assess the utility of the resulting images on accessible models, which are more likely to run on consumer-based systems.

---

> > > > ### Author Response · Authors · 2025-06-09
> > > >
> > > > We want to thank you again for engaging in the discussion period! We are also grateful for your decision to raise the score and for your positive assessment. We will gladly incorporate your final suggestions into the camera-ready version.

---

### Comment · Program_Chairs · 2025-04-03

This paper violates the page limit due to adding a limitation sections beyond the page limit. COLM does not have a special provision to allow for an additional page for the limitations section. However, due to this misunderstanding being widespread, the PCs decided to show leniency this year only. Reviewers and ACs are asked to ignore any limitation section content that is beyond the 9 page limit. Authors cannot refer reviewers to this content during the discussion period, and they are not to expect this content to be read.

---

### Author Response · Authors · 2025-06-03
**Response to General Feedback**

We thank the reviewers for their valuable feedback. Across the reviews, our work is recognized for addressing an important and socially impactful problem in assistive technologies (BRNH, MxCN), and for proposing a timely and technically sound framework (LRXX, BRNH). Reviewers also highlight the effectiveness of our approach in balancing privacy protection with utility (BRNH, MxCN), and note the value of the ecosystem risk score as an intuitive way to quantify privacy risks (LRXX).
___
We address general comments and questions below.

**+ The need for an agentic approach in this context**: We identify significant limitations of current SOTA models in accurately recognizing and localizing information present in challenging images. While it is possible that a non-agentic approach could also prove effective, we show that current non-agentric approaches do not perform well, and that our proposed agentic approach does. **Please see SOTA examples here: https://figpriv.github.io/**

Specifically:
- *Gemini 2.5 Flash Preview (05-20)* consistently produces completely unaligned masking.
- *GPT-4o* frequently fails to identify text, or masks only a few small, unaligned regions.
- *GPTo3*, despite long inference times (e.g., 10m to 7m 22s), is often unable to localize the text completely, or results in blanket-masking.
- *MistralOCR* does not output spatial localization and often struggles with text recognition.

Our multi-agent system achieves precise content localization, enabling the context-aware masking required for effective privacy preservation while maximizing content utility, critical for BLV users.

**+ Baselines Missing**: To the best of our knowledge, there is no prior work that addresses this issue. As mentioned, we run several experiments using proprietary models and paywall APIs, results available in this link: **https://figpriv.github.io**.

**+ Is the BIV-Priv-Seg dataset enough?**: To the best of our knowledge, the BIV-Priv-Seg dataset is the only real-world dataset that comprises images taken by BLV users and contains PII information. The objects in these images are often *occluded, zoomed in with content cropped out, blurry, in low lighting conditions, misaligned, among other challenges* (https://figpriv.github.io/bivpriv_samples.png). Moreover, all other splits from the VizWiz dataset have already been masked and PII removed.

Other existing datasets collected by BLV users are designed for outdoor navigation [1], scene understanding [2], or object recognition [3]; none of these datasets contains PII. We kindly ask the reviewer to help us identify other datasets with this type of information.

**+ Evaluation of utility feels suboptimal**: Given that we don’t have ground-truth questions and answers for the images, we measure whether the image content and text are **semantically aligned**, along with the **inter-annotator agreement** for the human annotations.

The evaluation results are obtained by computing the likelihood of the models to answer binary questions, which can be regarded as *confidence scores*, given that *the output is a probability of correctness*. We also include synthetic objects that are easily recognizable as a control set; the likelihood of recognizing the control object under varying masking conditions serves as an indicator of the model’s visual/language alignment. Table 3 shows that the models are more confident in identifying the control object when the private information is completely masked, lowering their confidence as the fine-grained and selective-fine-grained masking is applied.

We address each reviewer with detailed answers in threads below.
___________
[1] Islam, Md Touhidul, et al. "A Dataset for Crucial Object Recognition in Blind and Low-Vision Individuals' Navigation." arXiv preprint arXiv:2407.16777. ASSETS ‘24.

[2] Ruei-Che Chang, Yuxuan Liu, and Anhong Guo. 2024. WorldScribe: Towards Context-Aware Live Visual Descriptions. In Proceedings of the 37th Annual ACM Symposium on User Interface Software and Technology. UIST '24.

[3] Theodorou, L., Massiceti, D., Zintgraf, L. , Stumpf, S. , Morrison, C., Cutrell, E., Harris, M. T. & Hofmann, K. (2021). Disability-first Dataset Creation: Lessons from Constructing a Dataset for Teachable Object Recognition with Blind and Low Vision Data Collectors. In: The 23rd International ACM SIGACCESS Conference on Computers and Accessibility. ASSETS '21.

---

### Decision · Program_Chairs · 2025-07-08

**Decision:**

Accept

**Comment:**

The paper presents a new granular privacy protection framework, Fig-Priv, that aims to mask sensitive information in images.  Reviewers raised a few concerns, but they appear to have been addressed by the authors during the response period, and ultimately all reviewers recommend accepting the paper.